# GraphIC: A Graph-Based In-Context Example Retrieval Model for Multi-Step Reasoning

## Abstract

In-context learning (ICL) enables large language models (LLMs) to generalize to new tasks by incorporating a few in-context examples (ICEs) directly in the input, without updating parameters. However, the effectiveness of ICL heavily relies on the selection of ICEs, and conventional text-based embedding methods are often inadequate for tasks that require multi-step reasoning, such as mathematical and logical problem solving. This is due to the bias introduced by shallow semantic similarities that fail to capture the deeper reasoning structures required for these tasks. We present GraphIC, a novel approach that leverages graph-based representations of reasoning processes, coupled with Bayesian Networks (BNs) to select ICEs. Graph structures inherently filter out shallow semantics while preserving the core reasoning structure. Importantly, BNs capture the dependency of a node's attributes on its parent nodes, closely mirroring the hierarchical nature of human cognition—where each thought is shaped by preceding ones. This makes BNs particularly well-suited for multi-step reasoning tasks, aligning the process more closely with human-like reasoning. Extensive experiments across three types of reasoning tasks (mathematical reasoning, code generation, and logical reasoning) demonstrate that GraphIC outperforms both training-free and training-based models in selecting ICEs, excelling in terms of both effectiveness and efficiency. We show that GraphIC enhances ICL's performance and interpretability, significantly advancing ICE selection for multi-step reasoning tasks.

## 1 Introduction

In-context learning (ICL) (Brown et al., 2020) represents a paradigm in how large language models (LLMs) perform inference by using a small number of in-context examples (ICEs) within the input prompt. This technique enables LLMs to generalize to new tasks or enhance their performance on existing tasks without updating parameters. However, previous studies have highlighted the sensitivity of ICL performance to the specific ICEs selected (Zhao et al., 2021; Liu et al., 2022), underscoring the importance of strategic ICE selection. Consequently, numerous methods have been proposed to optimize the selection of ICEs, focusing on improving task performance and ensuring greater robustness (Liu et al., 2022; Rubin et al., 2022; Ye et al., 2023; Gupta et al., 2024). These methods frequently rely on text-based embeddings, where both the query and candidate ICEs are embedded using a language encoder, with similarity scores guiding the selection process.

Current text-based embedding selection methods primarily focus on capturing semantic-level similarity, demonstrating their utility in tasks such as semantic analysis (Liu et al., 2022) and machine translation (Agrawal et al., 2023). However, these approaches encounter significant limitations in multi-step mathematical and logical reasoning tasks, such as GSM8K (Cobbe et al., 2021) and ProofWriter (Tafjord et al., 2021). The core issue lies in the fact that textual data often encodes a substantial amount of shallow semantic information, which is largely irrelevant to the underlying reasoning processes required for math and logic tasks. This extraneous information introduces bias in the selection of ICEs (An et al., 2023), and can even lead the LLM to adopt misleading reasoning strategies, thereby degrading task performance. For example, in a problem involving speed calculation (i.e., determining the rate of change of distance over time), text-based embeddings may prioritize examples that focus on solving for time or distance due to their closer semantic similarity, as shown in Figure 1 (left). This misalignment steers the LLM away from the correct problem-

**Question:**

Marissa is hiking a 12-mile trail. She took 1 hour to walk the first 4 miles, then another hour to walk the next two miles. If she wants her average speed to be 4 miles per hour, what speed (in miles per hour) does she need to walk the remaining distance?

**Text-based Retrieval:**

Q: Jeannie hikes the 12 miles to Mount Overlook at a pace of 4 miles per hour, and then returns at a pace of 6 miles per hour. How long did her hike take, in hours?

A: Up, took 12 / 4 = 3 hours.
Down, took 12 / 6 = 2 hours.
Total time was 3 + 2 = 5 hours.

Q: Sadie, Ariana and Sarah are running a relay race. Each part of the race is a different length and covers different terrain. It takes Sadie 2 hours to run through the forest at an average speed of 3 miles per hour. Ariana sprints across the open field on her section at 6 miles per hour for half an hour. If Sarah runs along the beach at four miles per hour and their total time for the race is four and half hours, what is the total distance of the race?

A: Sadie ran for 3 miles/hour * 2 hour = 6 miles.
Ariana covered a distance of 6 miles/hour * (1/2) hour = 3 miles.
Sarah ran for 4.5 hours - 2 hours - 0.5 hours = 2 hours.
In this time, Sarah traveled a distance of 4 miles/hour * 2 hours = 8 miles.
The total distance for the race was 6 miles + 3 miles + 8 miles = 17 miles.
...

**Graph-based Retrieval:**

Q: Rachel is stuffing envelopes. She has eight hours to complete the task, and there are 1,500 envelopes. In the first hour, Rachel stuffs 135 envelopes. The second hour she stuffs 141 envelopes. How many envelopes will Rachel need to stuff per hour to finish the job?

A: Rachel has 1500 - 135 - 141 envelopes = 1224 envelopes remaining to stuff.
Rachel has 8 hours - 2 hours = 6 hours left to finish the task.
Rachel needs to stuff 1224 envelopes / 6 hours = 204 envelopes per hour.

Q: Allie has 9 toys, which are in total worth $52. If we know that one toy is worth $12, and all the other toys have the same value, how much does one of the other toys cost?

A: Allie has 9 - 1 = 8 toys of the same value.
Without the value of the one $12 toy, all 8 other toys are worth in total 52 - 12 = $40.
That would mean, that one of the other toys is worth 40 / 8 = $5.

...

**Output:**

12 miles / 4 mph = 3 hours
She has already walked for 1 + 1 + 1 = 3 hours
She has 3 - 3 = 0 hours left to walk the remaining distance
**She has 0 miles left to walk** ✗

**Output:**

Marissa has walked 4 + 2 = 6 miles so far.
She has 12 - 6 = 6 miles left to walk.
She wants to walk the entire trail in 12 miles / 4 mph = 3 hours.
She has already walked for 1 + 1 = 2 hours.
She has 3 - 2 = 1 hour left to walk the remaining 6 miles.
She needs to walk 6 miles / 1 hour = 6 mph. ✓

Figure 1: ICL with different ICE retrieval mechanisms. The left panel shows examples retrieved via BERT embedding (Devlin et al., 2019), while the right panel displays examples retrieved via GraphIC. Semantically related terms are highlighted in blue, and quantities or variables needing resolution are highlighted in green.

solving approach. Therefore, a novel representation is needed to enhance example retrieval for tasks involving multi-step mathematical and logical reasoning.

Extensive research indicates that graph-based representations, in contrast to text, align more closely with human cognitive processes and are better equipped to model multi-step reasoning (Friston, 2008; Besta et al., 2024; Yao et al., 2023). Graphs enable the filtering of irrelevant shallow semantic content while preserving the core reasoning structure, thus facilitating the development of an unbiased example retrieval mechanism. More critically, this representation offers a novel and interpretable approach for constructing example retrievers. When solving multi-step reasoning problems, the ideal examples are those whose reasoning processes can be directly applied to the query. In other words, after extracting the thought pattern from these examples, the same logic can be reused to solve the query question and arrive at the correct answer. Graph structures provide an explicit means to model these reasoning processes, enabling us to extract implicit thought patterns and assess their transferability to new problems.

More formally, a reasoning process can be modeled as a graph $G$ with attributes $(x_1, \ldots, x_n)$, where each $x_i$ represents the attribute of vertex $v_i$, corresponding to the thought at $v_i$. We further assume that the probability density function governing the sequence of thoughts throughout the reasoning process is given by $p(x_1, \ldots, x_n; G, W)$, where $W$ denotes the parameters of the underlying thought pattern. Given a query question $q$ and its associated reasoning graph $G^q$, our goal is to retrieve ICEs with parameterized thought patterns $W^i$ that maximize the probability density $p(x_1^q, \ldots, x_n^q; G^q, W^i)$, i.e., maximizing the likelihood of solving $q$ correctly. To achieve this, we employ a Bayesian Network (BN) (Pearl, 2014), a type of probabilistic graphical model, to represent the multi-step reasoning process and to parameterize the thought patterns. The motivation for using BN lies in their inherent similarity to human cognition; they assume that the value of each node (thought) is influenced by the values of preceding nodes, mirroring the way new ideas are constructed based on prior knowledge. This structure makes BNs particularly well-suited for capturing the dependencies and progression of reasoning steps.

In this paper, we introduce GraphIC, a **Graph**-based **In**-**C**ontext Example Retrieval Model. The key idea is to leverage graph representations of reasoning processes to enhance the retrieval of ICEs. Specifically, we first prompt the LLM to generate "thought graphs" for both the candidate examples

and the query question, where each graph encodes the reasoning steps involved. We then employ a BN to model each thought graph of candidate examples and estimate the associated parameters through maximum likelihood estimation, which we reformulate as a bilinear form maximization problem with a closed-form solution. To better simulate human-like reasoning, we incorporate a personalized PageRank (PPR) mechanism (Page, 1999), reflecting the cognitive tendency to revisit starting point during reasoning, a characteristic well-aligned with the assumptions behind PPR. Once the thought graphs and their parameters are computed, we select examples whose parameters maximize the probability density of thoughts on the query question's thought graph. Note that we leverage a graph model to aid ICE retrieval, not for direct problem-solving. The solution is driven by the LLM's reasoning capabilities. This approach ensures that the retrieved examples, though not necessarily semantically similar to the query, align with the underlying reasoning process. As shown in Figure 1(right), GraphIC retrieves examples that, while not focusing on semantically similar tasks, share a common reasoning structure—for instance, solving for rates such as how the number of envelopes changes over time or how costs vary with the number of toys. This guides the LLM towards the correct solution.

We evaluate GraphIC against 10 baseline models on three multi-step reasoning tasks: mathematical reasoning, code generation, and logical reasoning. Results show GraphIC excels in selecting relevant ICEs, surpassing existing methods in effectiveness and efficiency. Additionally, GraphIC shows faster performance improvements with more ICEs and exhibits asymmetry, mirroring real-world reasoning. Key contributions are summarized as follows: 1) We introduce a novel graph-based representation, the "thought graph", to model multi-step reasoning processes. This approach effectively filters out irrelevant shallow semantic information while preserving the essential reasoning steps. 2) By leveraging BNs, we design a retrieval mechanism that maximizes the probability density of solving the query problem, providing a more objective-driven and interpretable retrieval process. 3) Our experimental results indicate that GraphIC, despite being a training-free model, outperforms both training-free and training-based models across various multi-step reasoning tasks.

## 2 RELATED WORK

Existing ICE selection techniques can be classified as either training-free or training-based, depending on whether a retriever needs to be trained.

Training-free approaches are generally divided into two types: (i) those that use heuristic criteria such as similarity (Liu et al., 2022; Hu et al., 2022), diversity (Cho et al., 2023; Zhang et al., 2022b; Levy et al., 2023; Hongjin et al., 2022; Zhang et al., 2023), complexity (Fu et al., 2022), or combinations of these (Agrawal et al., 2023; Tonglet et al., 2023; Gupta et al., 2023) to select in-context examples (ICEs); (ii) those that leverage feedback from LLMs, such as probability distributions (Wu et al., 2023; Nguyen & Wong, 2023; Li & Qiu, 2023; Yang et al., 2023), perplexity (Gonen et al., 2023), or the model's generated output (An et al., 2023) to guide the selection process. While training-free approaches avoid the computational and time overhead associated with model training, their relatively simplistic architecture often results in sub-optimal performance compared to training-based methods.

Training-based methods are typically divided into two main categories. The first learns to select individual examples and then extends this to $k$-shot scenarios (Rubin et al., 2022; Xiong et al., 2024; Gupta et al., 2024). The second models the selection of a group of examples as a whole (Ye et al., 2023; Wang et al., 2023; Zhang et al., 2022a; Scarlatos & Lan, 2023; Lu et al., 2022; Peng et al., 2023; Xu et al., 2024). While training-based approaches usually achieve superior performance, their reliance on repeated LLM queries and model training makes them both computationally intensive and time-consuming.

Our proposed GraphIC method is not only training-free and inherently efficient but also incorporates an advanced graph-based example retriever specifically designed for multi-step reasoning tasks. This sophisticated design enables GraphIC to achieve a significant performance advantage, even surpassing training-based methods.

## 3 PRELIMINARIES: BAYESIAN NETWORK

A Bayesian Network (BN) (Pearl, 1982; 1986; PEARL, 1988; Heckerman et al., 1995; Friedman et al., 1997) is a probabilistic graphical model that represents conditional dependencies among random variables via a directed acyclic graph (DAG). In a DAG $G = (V, E)$, $V = \{v_1, \ldots, v_n\}$ denotes

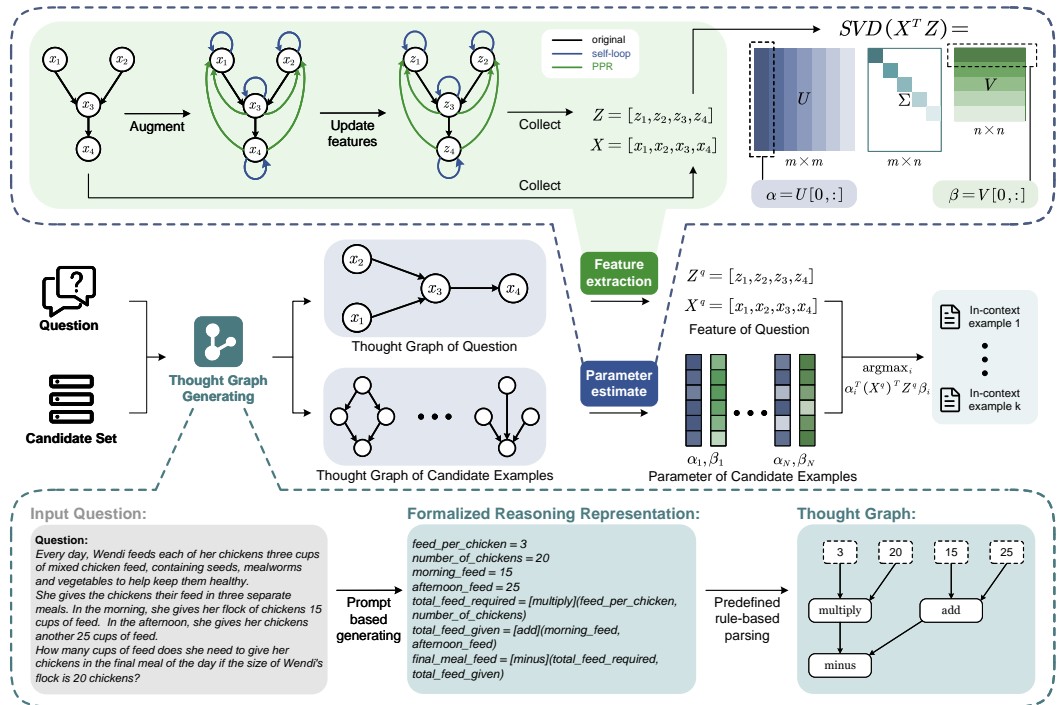

Figure 2: The overall pipeline of GraphIC. First, the question and candidate examples are processed through the thought graph generation module, where the LLM generates formalized reasoning representations, which are then parsed into thought graphs. For the question's thought graph, we extract $X$ (cognitive process per vertex) and compute aggregated feature $Z$. For candidates, parameters $\alpha_i$ and $\beta_i$ are estimated to capture relevant thought patterns. We then evaluate the applicability of these patterns on the query's thought graph, enabling ICE selection.

the vertices corresponding to random variables, and $E$ denotes the conditional dependencies. Each vertex $v_i$ is associated with a random variable $X_i$, and the joint probability distribution is factorized as:

$$p(x_1, x_2, \ldots, x_n) = \prod_{i=1}^{n} p(x_i | \text{pa}(v_i)), \qquad (1)$$

where $x_i \in \mathbb{R}^{n_f}$ denotes the value of the random variable $X_i$, $\text{pa}(v_i)$ refers to the set of parent variables for $v_i$, and $p(x_i | \text{pa}(v_i))$ is typically modeled as:

$$p(x_i | \text{pa}(v_i)) = g(\text{dist}(x_i, \hat{x}_i)), \qquad (2)$$

with $\hat{x}_i = W z_i$ and $z_i = f(\text{pa}(v_i))$. Here, $\hat{x}_i$ represents the predicted value of $x_i$ based on $z_i$, where $z_i$ aggregates information from the parent nodes $\text{pa}(v_i)$. The weight matrix $W$ is used to predict $\hat{x}_i$, $f(\cdot)$ denotes the aggregation function, $\text{dist}(\cdot, \cdot)$ is a distance metric between $x_i$ and $\hat{x}_i$, and $g(\cdot)$ is a function that satisfies: 1). monotonicity: $g'(u) \leq 0$ for $u \geq 0$; and 2) normalization: $\int_x g(\text{dist}(x, \hat{x}_i))dx = 1$.

Given the aggregated features $Z = (z_1, \ldots, z_n)^\top$ where $\top$ denotes the transpose operation, organizing the individual feature vectors $z_i$ into a matrix where each row corresponds to a feature vector, along with the distance function $\text{dist}(\cdot, \cdot)$ and function $g(\cdot)$, the joint probability density of the dataset $X = (x_1, x_2, \ldots, x_n)^\top$ can be computed.

## 4 THE PROPOSED GRAPHIC

In this work, we propose a novel approach called GraphIC for representing the problem-solving process through a graph-based model, intending to select examples that maximize the probability density of capturing the correct reasoning process. First, we introduce "thought graphs", a formal structure designed to represent the reasoning process underlying each example (Section 4.1). Building on this, we develop a probabilistic framework for modeling the thought graph, enabling us to compute the probability density associated with a given thought graph (Section 4.2). This probabilistic model serves as the foundation for our example selection strategy, which prioritizes examples

that maximize the probability density of thoughts on the thought graph of query question (Section 4.3). The overall framework is illustrated in Figure 2.

## 4.1 THOUGHT GRAPH AND ITS CONSTRUCTION

We begin by introducing the concept of a thought graph and its construction, inspired by the hierarchical structure of human cognition in problem-solving. The human thought process, especially when addressing complex problems, can be naturally modeled as a graph-like structure (Friston, 2008; Besta et al., 2024; Yao et al., 2023). In this work, we present the "thought graph" as a structure for modeling the cognitive process of LLMs during multi-step reasoning tasks. Formally, a thought graph $G$ is represented as a vertex-attributed graph, where each vertex is associated with a natural language text, corresponding to the description of the operation performed or the intermediate conclusion reached at that step. To facilitate computation, we further represent the vertex attributes as the BERT embedding (Devlin et al., 2019) of corresponding text, denoted as $x_i$.

Since LLMs are not natively equipped to output graph structures, we propose a methodology to generate these graphs from LLM outputs. As illustrated in Figure 2, we prompt the LLM to generate a "formalized reasoning representation" (FRR), which is subsequently parsed to construct the thought graph. The detailed prompt and parser pseudo-code are provided in Appendix D.

After constructing the thought graph, a straightforward way to select in-context examples (ICEs) is to compare the similarities between graph embeddings. To compute these embeddings, we employ a widely adopted method where each graph embedding is generated through iterative aggregation of node information, as outlined by Togninalli et al. (2019). Specifically, this process is formalized as:

$$X^{h+1} = \tilde{A}X^h, \ \ X^0 = X = (x_1, x_2, \ldots, x_n)^\top, \tag{3}$$

where $\tilde{A} = \tilde{D}_A^{-\frac{1}{2}}(A+I)\tilde{D}_A^{-\frac{1}{2}}$, $\tilde{D}_A = 1 + \sum_j A_{ij}$. $A$ represents the adjacency matrix of the thought graph, where $A_{ij} = 1$ indicates a directed edge from node $v_i$ to node $v_j$, and $A_{ij} = 0$ otherwise.

While this approach effectively captures the structural properties of human thought as represented in the graph, it is constrained by its focus on graph similarity alone. Importantly, selecting an example based solely on the similarity does not necessarily optimize the possibility of an LLM generating a correct reasoning trajectory. To overcome this, we further propose a novel example retrieval model that prioritizes the optimization of the probability density of producing a correct reasoning process detailed in the next subsection, moving beyond a mere reliance on graph similarity.

## 4.2 PROBABILISTIC MODEL ON THOUGHT GRAPH

Building on the method for constructing thought graphs, we now turn to developing a probabilistic model for this structure. BNs model the dependencies of a node's attributes on its parent nodes, which closely mirror the way human cognition functions—where new thoughts are informed by prior ones (Oaksford & Chater, 2007; Jacobs & Kruschke, 2011). This makes them a well-suited framework for modeling the thought graphs. In this section, we outline the construction of BNs for thought graphs. As described in Section 3, calculating the joint probability density on the thought graph requires the aggregate feature $Z$, the distance metric $\text{dist}(\cdot, \cdot)$, and the function $g(\cdot)$. We now provide a detailed discussion of how each of these components is constructed.

**Computing the Aggregated Feature $Z$.** Traditional BNs, which rely on the Markov assumption that a node's feature distribution depends solely on its parent nodes, are insufficient for modeling a thought graph where reasoning often requires referencing multiple prior steps. For example, problem solvers may need to iteratively review information from earlier steps or return to the beginning to re-examine the entire reasoning process. To address this limitation, we first employ an iterative aggregation mechanism that better captures the human reasoning processes. This iterative approach is formalized in Equation (3). Next, inspired by the Personalized PageRank (PPR) algorithm (Page, 1999; Gasteiger et al., 2018), we refine this method to more accurately simulate the flow of information during problem-solving. The PPR framework models a random walk where a user transitions between web pages with some probability of returning to the start. This closely parallels the cognitive process in complex problem-solving, where solvers often revisit initial hypotheses to reassess their reasoning. Therefore, the iterative feature aggregation is defined as follows:

$$X^{(h+1)} = \left[(1-\lambda)\tilde{A} + \lambda\tilde{B}\right]X^{(h)}, \ \ X^{(0)} = (x_1, x_2, \ldots, x_n)^\top, \tag{4}$$

where $\lambda \in (0,1)$, $\tilde{B} = \tilde{D}_B^{-\frac{1}{2}}(B+I)\tilde{D}_B^{-\frac{1}{2}}$, and $\tilde{D}_B = 1 + \sum_j B_{ij}$. The matrix $B$ models the retracing aspect of the thought process, where $B_{ij} = 1$ if $\deg(v_j) = 0$ and $\deg(v_i) > 0$, otherwise $B_{ij} = 0$, with $\deg(v_j)$ representing the in-degree of node $v_j$.

After $H$ iterations, the aggregated feature matrix $Z$ is given by:

$$Z = \left[(1-\lambda)\tilde{A} + \lambda\tilde{B}\right]^H X. \tag{5}$$

**Distance Metric dist$(\cdot)$ and Function $g(\cdot)$.** In prior works (Rubin et al., 2022; Ye et al., 2023; Xiong et al., 2024), the inner product has been a standard approach for quantifying vector similarity. Building on this, we define the distance function dist$(\cdot)$ (see equation 1) in terms of the inner production as follows:

$$\text{dist}(x_1, x_2) = l - x_1^\top x_2, \tag{6}$$

where $l$ is a sufficiently large constant chosen to ensure that dist$(x_1, x_2)$ remains positive. A suitable choice for $l$ is the square of the maximum norm of the embeddings produced by the model:

$$l = \max_t \left[\text{Emb}(t)^\top \text{Emb}(t)\right], \ t \in \mathcal{NL}, \tag{7}$$

with Emb$(\cdot)$ denoting a text embedding model and $\mathcal{NL}$ denoting the set of all natural languages. Additionally, we define $g_i(u) = \frac{1}{C_i}\exp(-u)$ (see equation 1), allowing us to represent:

$$p(x_i; G, X) = g_i(\text{dist}(x_i, \hat{x}_i)) = \frac{1}{C_i}\exp\left[-(l - \hat{x}_i^\top x_i)\right] = \frac{1}{C_i}\exp\left[-(l - z_i^\top W^\top x_i)\right], \tag{8}$$

where $C_i$ is a normalization constant.

Note that this formulation establishes a probabilistic model for the thought graph. Given the parameters $W$, we can compute the probability density of each vertex attribute, which in turn allows us to determine the probability density of the entire graph. In essence, the matrix $W$ governs the generation of new attributes and is meant to capture or represent the underlying structure of reasoning or connections between different concepts or ideas in the thought graph.

### 4.3 PROBABILISTIC EXAMPLE RETRIEVAL

As outlined in Section 4.2, the parameter $W$ is meant to capture and represent the underlying structure of reasoning or connections between different concepts or ideas within the thought graph. The task of computing the probability density of generating a particular thought graph given $W$ can be interpreted as evaluating the likelihood of producing the associated reasoning process based on the thought pattern encoded within $W$. Building on this idea, we design an example retrieval mechanism that estimates the model parameters for each candidate example and prioritizes those that maximize the probability density of the thought graph corresponding to the query. These selected examples serve as ICEs, offering the highest potential for accurately solving the problem at hand.

**Estimation of Model Parameters.** We estimate the parameter matrix $W$ by maximizing the likelihood function $\mathcal{L}_W$ of the thought graph features, which is computed as

$$\mathcal{L}_W = \prod_{i=1}^n p(x_i|G), \ \log\mathcal{L}_W = \sum_{i=1}^n \log p(x_i|G). \tag{9}$$

To simplify the computation, this reduces to the following:

$$\log\mathcal{L}_W = -\sum_{i=1}^n \log C_i + \sum_{i=1}^n \left[-(l - z_i^\top W^\top x_i)\right] = -\sum_{i=1}^n \log C_i - nl + \text{tr}(ZW^\top X^\top). \tag{10}$$

Hence, maximizing $\mathcal{L}_W$ is equivalent to maximizing $\text{tr}(ZW^\top X^\top)$, formally expressed as:

$$\max_W \ \text{tr}(ZW^\top X^\top), \ W \in \mathbb{R}^{n_f \times n_f}, \ \text{s.t.} \ ||W||_F = (\sum_{i=1}^m \sum_{j=1}^n w_{ij}^2)^{\frac{1}{2}} = 1. \tag{11}$$

This constraint ensures that the magnitude of $W$ does not influence the optimization.

Typically, the number of vertices $n$ in the thought graph is much smaller than the embedding dimensionality $n_f$ (i.e., $n \ll n_f$). For instance, in the GSM8K dataset, thought graphs often contain fewer than 20 vertices, while the embeddings $n_f$ can be as high as 768 if BERT is used. This dimensional disparity makes the solution for $W$ non-unique. Moreover, storing and computing a matrix of size $n_f \times n_f$ is computationally burdensome. To address both uniqueness and computational efficiency,

we constrain $W$ to to be of rank 1, reducing it to the form:

$$W = \alpha\beta^\top, \ ||\alpha||_2 = ||\beta||_2 = 1, \ \alpha, \beta \in \mathbb{R}^{n_f}. \tag{12}$$

This simplifies the optimization to:

$$\text{tr}(ZW^\top X^\top) = \text{tr}(Z\beta\alpha^\top X^\top) = \text{tr}(\alpha^\top X^\top Z\beta) = \alpha^\top X^\top Z\beta. \tag{13}$$

Thus, we reformulated the problem as a bilinear form maximization problem:

$$\max_{\alpha, \beta} \alpha^\top X^\top Z\beta, \quad \text{s.t.} \quad ||\alpha||_2 = ||\beta||_2 = 1. \tag{14}$$

The closed-form solution to this problem (Leon, 1994) can be obtain as:

$$\alpha = U[0, :], \ \beta = V[0, :], \ \text{where} \quad U, \Sigma, V = \text{SVD}(X^\top Z). \tag{15}$$

**Selection of Examples.** We extract parameters $(\alpha, \beta)$ from the thought graph of each candidate example, denoted as $\{(\alpha^1, \beta^1)\}_{i=1}^N$, where $N$ represents the size of the candidate set. For a given query $q$, we construct its thought graph $G^q$ and derive $Z^q$, then we select the top $k$ candidate examples that maximize the probability density $p_i(X^q)$ of generating the correct reasoning process:

$$p_i(X^q) = (\alpha^i)^\top (X^q)^\top Z^q \beta^i. \tag{16}$$

These selected examples are taken as ICEs in ICL, where the detailed templates used in ICL are given in Appendix B.

## 5 EXPERIMENTS

### 5.1 DATASETS AND IMPLEMENTATION DETAILS

We conduct a comprehensive evaluation of GraphIC model across four multi-step reasoning benchmarks: two for mathematical reasoning (GSM8K (Cobbe et al., 2021) and AQUA (Ling et al., 2017)), one for code generation (MBPP (Austin et al., 2021)), and one for logical reasoning (ProofWriter (Tafjord et al., 2021)). For both GSM8K and MBPP, we utilize the original datasets without further preprocessing. For AQUA and ProofWriter, we refine the original dataset to improve the experimental setup, as detailed in Appendix A.

For GSM8K, AQUA, and ProofWriter, model performance is evaluated based on the accuracy of the LLMs' final answers. For MBPP, we adopt the pass@1 metric (Chen et al., 2021) to assess the quality of code generation.

We employ GPT-4o-mini and Llama-3.1-8B-Instruct as LLMs. Unless explicitly mentioned otherwise, all evaluations are conducted under an 8-shot paradigm. We set the temperature to 1e-5. We set iterations $H$ in Equation 5 to 3, with $\lambda$ values from $\{0, 0.1, 0.2, 0.3\}$, based on the LLM and dataset (see Appendix D for details). For GSM8K, AQUA, and ProofWriter, we prompt the LLM to create a formalized reasoning representation (FRR) for thought graph construction, using vertex features from a BERT model. For MBPP, we use the `staticfg` module to parse Python code and generate the control flow graph, embedding each vertex's features with CodeBERT (Feng et al., 2020). Variable names in the code are anonymized with arbitrary symbols like 'a', 'b', and 'c'.

### 5.2 BASELINES

Our model, GraphIC, is designed as a training-free retriever for ICE selection. We compare GraphIC against six training-free retrieval methods spanning random, similarity-based, diversity-based, and complexity-based approaches, including: 1) **Random** randomly selects $k$ unique ICEs from the candidate set; 2) **BM25** (Robertson et al., 2009) selects the top $k$ examples based on BM25 scoring; 3) **BERT** (Devlin et al., 2019) is a dense retriever using cosine similarity with BERT-base-uncased embeddings; 4) **Complex-CoT** (Fu et al., 2022) selects $k$ examples based on complexity, quantified by newline characters; 5) **Auto-CoT** (Zhang et al., 2022b) clusters candidates and selects the closests to each cluster center; and 6) **Skill-kNN** (An et al., 2023) prompts LLM to generate task-relevant skills for query and candidates, followed by dense retrieval. Since Skill-kNN does not natively support datasets like GSM8K, we manually craft the instructions and examples, detailed in Appendix C.

We also compare with four training-based retrievers, which encompass both single-example and combination-example retrieval strategies, including: 1) **EPR** (Rubin et al., 2022) is trained to retrieve the single most relevant ICE, with top $k$ examples being selected during inference; 2) **CEIL** (Ye et al., 2023) uses determinantal point processes to select ICEs balancing similarity and diversity, where three CEIL models per dataset with scaling factors of 0.01, 0.05, and 0.1 are trained and the best results are reported; 3) **DQ-LoRe** (Xiong et al., 2024) uses dual queries and

low-rank approximation re-ranking to identify ICEs; and 4) **GistScore** (Gupta et al., 2024) encodes task-specific information into gist tokens for selecting ICEs. Following Skill-kNN, We use GPT-J-6B (Wang & Komatsuzaki, 2021) as the scoring LLM for EPR, CEIL, and DR-LoRe.

## 5.3 MAIN RESULTS

Figure 3 illustrates the thought graphs corresponding to each dataset. Table 1 evaluates our GraphIC model against 10 baselines across two LLMs and four datasets. As a training-free method, GraphIC consistently outperforms both training-free and training-based baselines in most settings. With the GPT-4o-mini model, GraphIC achieves the highest performance, averaging 2.57% above the leading training-free model and 1.18% above the best training-based model. For the Llama-3.1-8B-Instruct model, GraphIC ranks first in three out of four datasets, with an average gain of 4.29% over the top training-free competitor and 2.5% over the strongest training-based method. Our analysis shows that the GraphIC model significantly enhances performance in mathematical and logical reasoning tasks versus code generation, especially for complex problems. For instance, in the GSM8K dataset, GraphIC outperforms all baselines by an average of 0.65% and 3.57% with two LLMs. In the more challenging AQUA dataset, improvements rise to 3.47% and 7.64%.

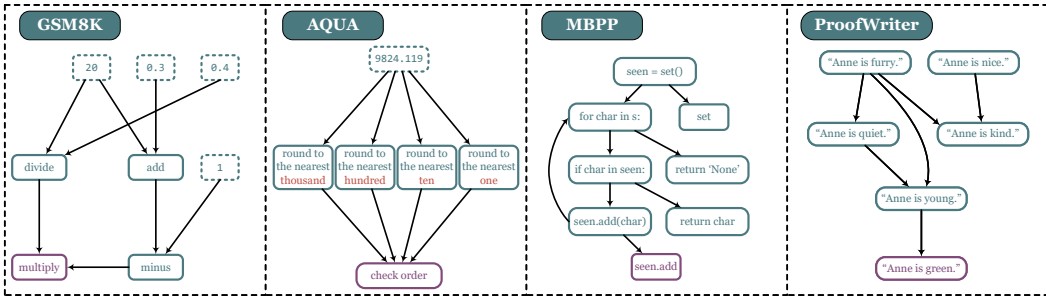

Figure 3: Examples of thought graphs. For GSM8K and AQUA, Vertices indicated by dashed lines represent the numbers entered during calculations, which will be removed in subsequent steps. The purple vertice indicates the final step in the reasoning process.

Table 1: Main results on two LLMs and four datasets. For random retrieval, we present the mean and standard deviation derived from five independent experiments. **Bold numbers** indicate the best results, while underlined numbers represent the second-best results.

| LLM | Model | GSM8K | AQUA | MBPP | ProofWriter | Avg. |
|---|---|---|---|---|---|---|
| GPT-4o-mini | Random | 92.90 (0.31) | 71.58 (0.72) | 72.76 (0.74) | 64.90 (0.93) | 75.54 (0.36) |
| | BM25 | 92.64 | 70.47 | 73.4 | 66.25 | 75.69 |
| | BERT | 93.02 | 66.93 | 74.2 | 65.25 | 74.85 |
| | Complex-CoT | 92.49 | 67.32 | 74.2 | 64.25 | 74.57 |
| | Auto-CoT | 92.72 | 69.69 | 73.8 | 62.25 | 74.62 |
| | Skill-kNN | 92.34 | 71.65 | 72.0 | 66.00 | 75.50 |
| | EPR | 93.02 | 72.04 | 73.8 | 68.50 | 76.84 |
| | CEIL | 92.57 | 72.44 | 73.8 | 69.50 | 77.08 |
| | DQ-LoRe | 93.32 | 69.69 | 74.6 | 66.50 | 76.03 |
| | GistScore | 93.25 | 69.69 | 72.8 | 67.00 | 75.69 |
| | GraphIC | **93.48** | **73.62** | **75.2** | **70.75** | **78.26** |
| Llama-3.1 -8B-Instruct | w/o ICL | 46.47 | 35.43 | 43.4 | 40.75 | 41.51 |
| | Random | 78.86 (0.87) | 53.15 (1.85) | 57.72 (1.06) | 76.10 (2.45) | 66.46 (0.84) |
| | BM25 | 77.71 | 46.85 | 62.0 | 77.75 | 66.08 |
| | BERT | 74.15 | 50.39 | 60.8 | 73.75 | 64.77 |
| | Complex-CoT | 79.30 | 50.00 | 58.6 | 78.25 | 66.54 |
| | Auto-CoT | 72.78 | 42.91 | 58.4 | 78.00 | 63.02 |
| | Skill-kNN | 77.56 | 50.39 | 60.8 | 74.00 | 65.69 |
| | EPR | 75.66 | 53.94 | 62.0 | 79.25 | 67.71 |
| | CEIL | 75.51 | 51.97 | **62.4** | 81.00 | 67.72 |
| | DQ-LoRe | 77.93 | 54.33 | 59.8 | 81.25 | 68.33 |
| | GistScore | 74.60 | 44.49 | 60.4 | 79.50 | 64.75 |
| | GraphIC | **79.98** | **57.48** | 61.6 | **84.25** | **70.83** |

Additionally, we observe that the GPT-4o-mini model's performance on the GSM8K dataset is relatively invariant to the selection of ICEs. So, we further perform an additional experiment on the

Table 2: Results obtained using GPT-3.5-Turbo as the LLM on the GSM8K dataset.

| Random | BM25 | BERT | Complex-CoT | Auto-CoT | Skill-kNN | EPR | CEIL | DQ-LoRe | GistScore | GraphIC |
|---|---|---|---|---|---|---|---|---|---|---|
| 80.76(0.55) | 82.10 | 80.89 | 81.65 | 82.03 | 81.50 | 81.65 | 81.72 | 82.10 | 81.72 | **82.79** |

GSM8K dataset using the GPT-3.5-Turbo model. As presented in Table 2, GraphIC model achieves superior performance across all metrics.

## 5.4 Ablation Study

We perform a series of ablation studies to systematically evaluate the contribution of each component within the GraphIC framework, which is built upon three key pillars: the incorporation of thought graphs, PPR for aggregating features, and BN-based retrieval.

To this end, we develop several variants of the GraphIC model: 1) **Text** relies solely on text embeddings, the same as the BERT approach; 2) **FRR** retrieves examples using BERT embeddings derived from FRRs (or CodeBERT embeddings for the MBPP dataset); 3) **Graph** utilizes the formula (3) to generate graph embeddings, which are employed for dense retrieval; 4) **Graph+PPR** uses the formula (5) to obtain graph embeddings for dense retrieval; 5) **Graph+BN** excludes the backtracking (or PPR) mechanism from the full GraphIC model during computing $Z$; and 6) **Graph+PPR+BN** represents the full GraphIC model, integrating all components.

We conduct experiments leveraging the Llama-3.1-8B-Instruct model across four datasets, with the outcomes detailed in Table 3. The findings underscore that each component of GraphIC plays a pivotal role in boosting model performance, with the most significant improvements observed when all three components are utilized in conjunction.

Table 3: Ablation Study.

| Model | GSM8K | AQUA | MBPP | ProofWriter |
|---|---|---|---|---|
| Text | 74.15 | 50.39 | 60.8 | 73.75 |
| FRR | 78.31 | 50.78 | 60.4 | 82.50 |
| Graph | 78.46 | 54.72 | 60.4 | 83.50 |
| +PPR | 78.92 | 56.30 | 61.0 | 83.75 |
| +BN | 79.07 | 49.21 | 60.4 | **84.25** |
| +PPR+BN | **79.98** | **57.48** | **61.6** | **84.25** |

## 5.5 Analysis

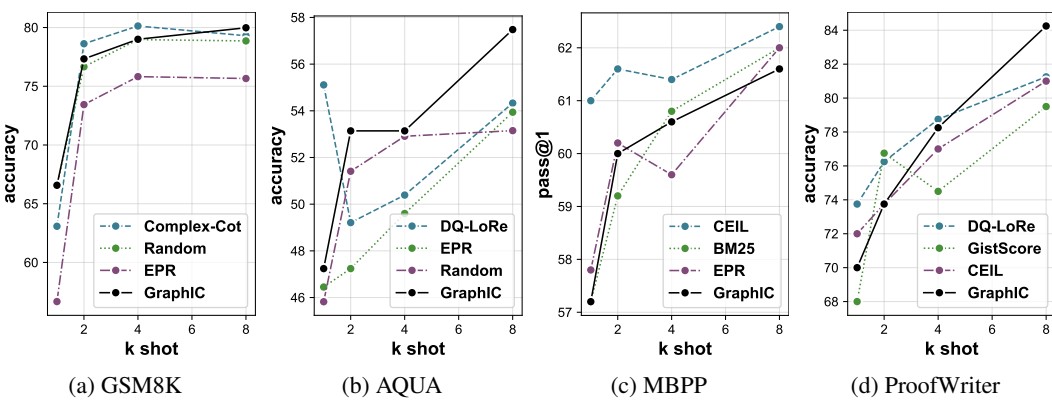

(a) GSM8K     (b) AQUA     (c) MBPP     (d) ProofWriter

Figure 4: Comparison of different numbers of ICEs on various datasets. The blue, red, and purple lines indicate top 1, top 2, and top 3 performing baseline as shown in Table 1, respectively.

**Impact of ICE Examples on Model Performance.** We conduct an in-depth investigation into the influence of the number of ICEs on the performance of our proposed GraphIC model and several competitive baselines across four datasets. For each dataset, we select the top three baselines and varied the number of examples in the set {1, 2, 4, 8}. Llama-3.1-8B-Instruct is employed as the underlying LLM. Results in Figure 4 indicate a general trend of improved model performance with an increase in the number of examples. Notably, The performance of GraphIC steadily improves as the number of ICEs increases, unlike some baseline methods, which may experience performance degradation when the number of ICEs increases. Furthermore, while GraphIC initially lags behind the baselines in the low-shot settings, its performance exhibits a more pronounced improvement as the number of examples grew. One can observe that GraphIC surpasses the baselines, demonstrating superior performance and underscoring its robustness as the number of examples increases.

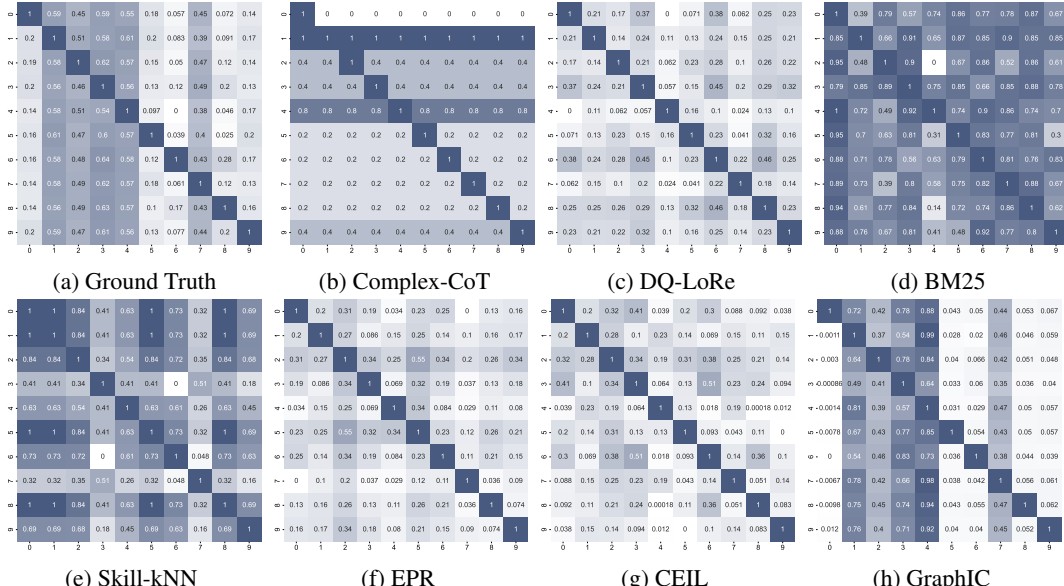

Figure 5: Ground truth matrix and score matrices of various models. The matrix values have been linearly scaled to the range [0,1], and the diagonal elements have been set to 1.

**Assumption of Symmetry.** Our findings show that the GraphIC model uses an "asymmetric" approach, unlike the common symmetry assumption in most baseline models. To assess the validity of this assumption, we conduct an experiment examining whether symmetry, in the context of retrieval models, holds. Here, a retrieval model is considered symmetric if $\text{score}(i, j) = \text{score}(j, i)$, where $\text{score}(i, j)$ represents the model's assessment of example $i$ as an ICE for example $j$. For example, the Skill-kNN is symmetric as it uses inner product embeddings for $\text{score}(j, i)$, while the Complex-CoT model is asymmetric, calculating $\text{score}(j, i)$ based on the complexity of example $i$.

We randomly select 10 examples from the GSM8K candidate set and use the top-7 performing models to compute the score matrix $S$ ($S_{ij} = \text{score}(i, j)$), which we then compare against the ground truth matrix $S^{\text{gt}}$. Here, $S_{ij}^{\text{gt}}$ captures the probability that Llama-3.1-8B-Instruct provides the correct answer when example $i$ is used as an ICE for example $j$.

The experimental results show that the ground truth matrix (Figure 5 (a)) is asymmetric, undermining the symmetry assumption. Using symmetric models for inherently asymmetric data introduces significant errors. For instance, the EPR model, trained on correct answer probabilities, struggles with accuracy due to its symmetry reliance (Figure 5 (f)). In contrast, simple asymmetric methods like Complex-CoT (Figure 5 (b)) and BM25 (Figure 5 (d)) perform well, ranking second and fourth as Table 1 shows, and surpassing many symmetric models. However, their simplistic assumptions limit their ability to capture ground truth nuances. In contrast, GraphIC (Figure 5 (h)), a sophisticated asymmetric model, aligns closely with the ground truth, resulting in superior performance.

## 6 CONCLUSION

We introduce GraphIC, a graph-based method for in-context example (ICE) retrieval aimed at enhancing LLM performance on multi-step reasoning tasks. By modeling reasoning as "thought graphs" and utilizing Bayesian Networks and personalized PageRank, GraphIC selects ICEs that align with the task's cognitive structure, overcoming the limitations of text-based embedding methods. Extensive experiments on four benchmarks show that GraphIC consistently outperforms both training-free and training-based baselines, especially in mathematical and logical reasoning. Our analysis of symmetry assumptions highlights the advantage of asymmetric retrieval models. A limitation of the GraphIC model is that, as a training-free framework, it may face difficulties in capturing more intricate thought patterns. Beyond this, GraphIC not only introduces a powerful ICEs retrieval method, but more crucially, it provides a way to represent and understand the reasoning process. This capability can be applied to various domains related to LLM reasoning, such as developing novel graph-based reasoning methods, selecting high-quality and diverse training datasets, and more.

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

# A PROCESSING OF AQUA AND PROOFWRITER

Given the substantial size of the AQUA dataset, which incurs significant retrieval overhead during testing, we followed the methodology outlined in DQ-LoRe (Xiong et al., 2024), using a 1,000-sample subset for efficient evaluation.

For the ProofWriter dataset, we refined the subset selected by Logic-LM (Pan et al., 2023), excluding instances labeled as "Unknown," as these samples lacked explicit reasoning chains. Furthermore, because the original training set did not provide reasoning in natural language, we leveraged the GPT-4o-mini model to generate reasoning sequences for the training set, discarding any generated outputs deemed incorrect. We evaluate the correctness of the reasoning process by the correctness of the final result, which is a commonly used approach (Lightman et al., 2024; Xiong et al., 2024; Khattab et al., 2022). This process resulted in a refined training set of 1,358 examples with their Chains of Thought and 400 test samples from the original ProofWriter dataset.

# B PROMPT TEMPLATES

For the four datasets under consideration, we design the following prompt templates to format the ICEs and the question into a prompt, which is then fed into an LLM to generate answers.

GSM8K & AQUA:

```
Q: {{ice_question_1}}

A: {{ice_answer_1}}

...

Q: {{ice_question_k}}

A: {{ice_answer_k}}

Q: {{question}}

A:
```

MBPP:

```
Text: {{ice_question_1}}

Test Cases: {{ice_test_cases_1}}

Code: {{ice_code_1}}

...

Text: {{ice_question_k}}

Test Cases: {{ice_test_cases_k}}

Code: {{ice_code_k}}

Text: {{question}}

Test Cases: {{test_cases}}

Code:
```

ProofWriter:

```
Q: {{ice_question_1}}

Proof: {{ice_answer_1}}

...

Q: {{ice_question_k}}

Proof: {{ice_answer_k}}

Q: {{question}}
```

```
Proof:
```

## C   SKILL-κNN

Since Skill-kNN does not offer prompts for skill generation in these three tasks, we referred to the prompt designed for the semantic parsing task in the original paper to write prompts for the four datasets we used. First, we applied the Complex-CoT method to select 8 examples, then employed the GPT-4o model to generate skills in a zero-shot setting. Finally, we integrated these results to construct the final prompt.

GSM8K:

```
Generate the skills needed to solve the following math problems.

Q: You can buy 4 apples or 1 watermelon for the same price. You bought 36 fruits evenly split
    between oranges, apples and watermelons, and the price of 1 orange is $0.50. How much
    does 1 apple cost if your total bill was $66?

Skills:
1. Algebraic Reasoning
2. Proportional Thinking
3. Numerical Operations
4. Logical Analysis
5. Problem Solving
6. Cost Analysis

...

Q: {{question}}

Skills:
```

AQUA:

```
Generate the skills needed to solve the following math problems.

Q: In a group of 6 boys and 4 girls, four children are to be selected. In how many different
    ways can they be selected such that at least one boy should be there?
Options: A)209, B)210, C)211, D)213, E)215

Skills:
1. Selection Principles
2. Inclusion-Exclusion
3. Logical Analysis
4. Quantitative Reasoning

...

Q: {{question}}

Skills:
```

MBPP:

```
Generate the skills needed to solve the following coding problems.

Text: Write a function to generate a square matrix filled with elements from 1 to n raised to
    the power of 2 in spiral order.

Test Cases:
assert generate_matrix(3)==[[1, 2, 3], [8, 9, 4], [7, 6, 5]]
assert generate_matrix(2)==[[1,2],[4,3]]
assert generate_matrix(7)==[[1, 2, 3, 4, 5, 6, 7], [24, 25, 26, 27, 28, 29, 8], [23, 40, 41,
    42, 43, 30, 9], [22, 39, 48, 49, 44, 31, 10], [21, 38, 47, 46, 45, 32, 11], [20, 37, 36,
    35, 34, 33, 12], [19, 18, 17, 16, 15, 14, 13]]

Skills:
1. Matrix Manipulation
2. Spiral Algorithm Design
3. Loop Control Flow
4. Boundary Handling
5. Efficient Implementation
6. Testing & Debugging
7. Sequence-to-Matrix Mapping
```

```
...

Text: {{question}}

Test Cases: {{test_cases}}

Skills:
```

ProofWriter:

```
Generate the skills needed to solve the following logical reasoning problems.

Q: Triples:
1. Anne is not big.
2. Anne is cold.
3. Anne is red.
4. Dave is green.
5. Dave is rough.
6. Erin is green.
7. Erin is kind.
8. Erin is rough.
9. Fiona is green.
10. Fiona is not nice.
Rules:
1. If Erin is cold then Erin is rough.
2. If something is rough then it is nice.
3. All green, big things are kind.
4. If Dave is kind then Dave is cold.
5. If something is green and not rough then it is big.
6. All nice, rough things are big.
7. If Dave is cold and Dave is nice then Dave is red.
Based on the above information, is the following statement true or false? Dave is red.
A) True B) False

Skills:
1. Comprehension
2. Logical Deduction
3. Conditional Reasoning
4. Contrapositive Reasoning
5. Transitive Reasoning
6. Identify Necessary Conditions
7. Eliminate Contradictions
8. Pattern Recognition
9. Attention to Detail
10. Inference Making

...

Q: {{question}}

Skills:
```

# D GRAPHIC

## D.1 FORMALIZED REASONING REPRESENTATION

The prompt examples below are used to generate formalized reasoning representations for the four datasets being considered. For the test question, since no answer is provided, we will remove the section of the prompt highlighted in blue. This will allow the LLM to generate both the answer and the formalized reasoning representation simultaneously, from which we can then extract the formalized reasoning representation.

GSM8K:

```
Translate the given calculations into code form. Each line of code MUST follow the format
    specified below:
output_variable = [description of operation](input_variable_1, ..., input_variable_n)

Q: You can buy 4 apples or 1 watermelon for the same price. You bought 36 fruits evenly split
    between oranges, apples and watermelons, and the price of 1 orange is $0.50. How much
    does 1 apple cost if your total bill was $66?

A: If 36 fruits were evenly split between 3 types of fruits, then I bought 36/3 =
    <<36/3=12>>12 units of each
```

```
fruit
If 1 orange costs $0.50 then 12 oranges will cost $0.50 * 12 = $<<0.5*12=6>>6
If my total bill was $66 and I spent $6 on oranges then I spent $66 - $6 = $<<66-6=60>>60 on
    the other 2 fruit types.
Assuming the price of watermelon is W, and knowing that you can buy 4 apples for the same
    price and that the price
of one apple is A, then 1W=4A
If we know we bought 12 watermelons and 12 apples for $60, then we know that $60 = 12W + 12A
Knowing that 1W=4A, then we can convert the above to $60 = 12(4A) + 12A
$60 = 48A + 12A
$60 = <<60=60>>60A
Then we know the price of one apple (A) is $60/60= $<<60/60=1>>1
#### 1

Code:
total_fruits = 36
types_of_fruits = 3
price_per_orange = 0.50
total_oranges = 12
total_bill = 66
equivalent_apples_for_watermelon = 4
total_apples_and_watermelons = 12
fruits_per_type = [divide](total_fruits, types_of_fruits)
cost_of_oranges = [multiply](total_oranges, price_per_orange)
remaining_budget = [minus](total_bill, cost_of_oranges)
price_per_apple = [construct and solve an equation](total_apples_and_watermelons,
    equivalent_apples_for_watermelon, remaining_budget)

...

Q: {{question}}

A: {{answer}}

Code:
```

AQUA:

```
Translate the given calculations into code form. Each line of code MUST follow the format
    specified below:
output_variable = [description of operation](input_variable_1, ..., input_variable_n)

Q: In a group of 6 boys and 4 girls, four children are to be selected. In how many different
    ways can they be selected such that at least one boy should be there?
Options: A)209, B)210, C)211, D)213, E)215

A: To determine the number of ways to select 4 children from a group of 6 boys and 4 girls
    such that at least one boy is included, we will use the method of complement counting.

First, let's calculate the total number of ways to select 4 children from 10 children (6 boys
    + 4 girls):

\[
\binom{10}{4} = \frac{10!}{4!(10-4)!} = \frac{10 \times 9 \times 8 \times 7}{4 \times 3 \times
    2 \times 1} = 210
\]

Next, we calculate the number of ways to select 4 children with no boys, i.e., all girls.
    Since there are only 4 girls, and we need to select all 4 of them:

\[
\binom{4}{4} = 1
\]

Now, subtract the number of ways to select all girls from the total number of ways to select 4
    children to find the number of ways that include at least one boy:

\[
\binom{10}{4} - \binom{4}{4} = 210 - 1 = 209
\]

Thus, the number of ways to select 4 children with at least one boy is:

\[
\boxed{209}
\]
#### A

Code:
total_children = 10
```

```
children_to_select = 4
boys = 6
girls = 4
total_ways_to_select = [combination](total_children, children_to_select)
all_girls_selection = [combination](girls, children_to_select)
ways_with_at_least_one_boy = [subtract](total_ways_to_select, all_girls_selection)

...

Q: {{question}}

A: {{answer}}

Code:
```

MBPP:

```
Text: Write a function to generate a square matrix filled with elements from 1 to n raised to
    the power of 2 in spiral order.

Test Cases:
assert generate_matrix(3)==[[1, 2, 3], [8, 9, 4], [7, 6, 5]]
assert generate_matrix(2)==[[1,2],[4,3]]
assert generate_matrix(7)==[[1, 2, 3, 4, 5, 6, 7], [24, 25, 26, 27, 28, 29, 8], [23, 40, 41,
    42, 43, 30, 9], [22, 39, 48, 49, 44, 31, 10], [21, 38, 47, 46, 45, 32, 11], [20, 37, 36,
    35, 34, 33, 12], [19, 18, 17, 16, 15, 14, 13]]

Code:
def generate_matrix(n):
    if n<=0:
        return []
    matrix=[row[:] for row in [[0]*n]*n]
    row_st=0
    row_ed=n-1
    col_st=0
    col_ed=n-1
    current=1
    while (True):
        if current>n*n:
            break
        for c in range (col_st, col_ed+1):
            matrix[row_st][c]=current
            current+=1
        row_st+=1
        for r in range (row_st, row_ed+1):
            matrix[r][col_ed]=current
            current+=1
        col_ed-=1
        for c in range (col_ed, col_st-1, -1):
            matrix[row_ed][c]=current
            current+=1
        row_ed-=1
        for r in range (row_ed, row_st-1, -1):
            matrix[r][col_st]=current
            current+=1
        col_st+=1
    return matrix

...

Text: {{question}}

Test Cases:
{{test_list}}

Code:
```

ProofWriter:

```
Formalize the following proof process based on the given information. Where Each line of
    formalized proof MUST follow the format specified below.
output_triple = [rule used](input_triple_1, ... , input_triple_n)

Q: Triples:
1. Anne is not big.
2. Anne is cold.
3. Anne is red.
4. Dave is green.
5. Dave is rough.
```

```
6. Erin is green.
7. Erin is kind.
8. Erin is rough.
9. Fiona is green.
10. Fiona is not nice.
Rules:
1. If Erin is cold then Erin is rough.
2. If something is rough then it is nice.
3. All green, big things are kind.
4. If Dave is kind then Dave is cold.
5. If something is green and not rough then it is big.
6. All nice, rough things are big.
7. If Dave is cold and Dave is nice then Dave is red.
Based on the above information, is the following statement true or false? Dave is red.
A) True B) False

Proof: To determine if the statement "Dave is red" is true or false, let's evaluate the given
    information step by step:

From the triples:
1. Dave is green.
2. Dave is rough.

From the rules:
1. If something is rough then it is nice. (Dave is rough, so Dave is nice.)
2. If Dave is cold and Dave is nice then Dave is red. (However, there's no information saying
    Dave is cold, so we can't use this rule directly.)
3. All nice, rough things are big. (Dave is nice and rough, so Dave is big.)

Now, we need to verify whether Dave being cold can be derived:
1. If Dave is kind then Dave is cold. (To use this, we need to verify if Dave is kind.)
   - All green, big things are kind. (Dave is green and big, so Dave is kind.)
   - Hence, Dave is kind, which according to rule 4, makes Dave cold.

Since now we know:
- Dave is cold.
- Dave is nice.

We can use rule 7:
- If Dave is cold and Dave is nice then Dave is red.

Since both conditions are satisfied, Dave is red.

The statement "Dave is red" is **True (A)**.

Formalized Proof:
"Dave is nice." = [rule2]("Dave is rough.")
"Dave is big." = [rule6]("Dave is nice.", "Dave is rough.")
"Dave is kind." = [rule3]("Dave is green.", "Dave is big.")
"Dave is cold." = [rule4]("Dave is kind.")
"Dave is red." = [rule7]("Dave is cold.", "Dave is nice.")

...

Proof: {{answer}}

Formalized Proof:
```

The pseudo-code of a parser that transforms formalized reasoning representations into a thought graph is provided in Algorithm 1. *Inputs*, *Output*, and *OperationName* are extracted following the pattern outlined below.

```
Output = [OperationName](input_1, ..., input_n)
```

## D.2 VALUES OF $\lambda$

We select hyper parameter $\lambda$ values from $\{0, 0.1, 0.2, 0.3\}$, and report the $\lambda$ values chosen on various datasets and LLMs in Table 4.

---

**Algorithm 1** Parsing Formalized Reasoning Representation

---

**Require:** formalized reasoning representation $FRR$
**Ensure:** Corresponding graph $G(V, E)$

1: $NodeSet \leftarrow \emptyset$
2: $EdgeSet \leftarrow \emptyset$
3: $line \leftarrow$ first line of $FRR$
4: **while** $line \neq$ NULL **do**
5:      Extract *Inputs*, *Output*, and *OperationName* from $line$
6:      **for** each $input$ in *Inputs* **do**
7:          **if** $input \notin NodeSet$ **then**
8:              Add $input$ to $V$
9:          **end if**
10:     **end for**
11:     **if** $Output \notin V$ **then**
12:        Add $Output$ to $V$, labeled as *OperationName*
13:     **end if**
14:     **for** each $input$ in *Inputs* **do**
15:        Add directed edge from $input$ to $Output$ to $E$
16:     **end for**
17:     $line \leftarrow$ next line of $FRR$
18: **end while**
19: $G = G(V, E)$

---

Table 4: $\lambda$ values chosen on various datasets and LLMs.

| Engine | GSM8K | AQUA | MBPP | ProofWriter |
|---|---|---|---|---|
| GPT-4o-mini | 0.2 | 0.2 | 0.1 | 0.1 |
| Llama-3.1-8B-Instruct | 0.3 | 0.2 | 0.2 | 0.0 |
| GPT-3.5-Turbo | 0.3 | / | / | / |

# E   SUPPLEMENTARY EXPERIMENTS

## E.1   CORRECTNESS OF LLM GENERATED ANSWERS FOR CREATING THOUGHT GRAPHS

To analyze the consistency between LLM-generated answers and real solutions, we tested the accuracy of these answers used to generate the thought graphs. The results are shown in the Table 5. From Table 5, it can be seen that these answers used to generate the thought graph already have relatively high accuracy, which ensures their consistency with the real solution. Furthermore, the table demonstrates that using the thought graph to retrieve examples can further improve accuracy, especially in mathematical reasoning and logical reasoning tasks. We use Llama-3.1-8B-Instruct for testing.

Table 5: Correctness of LLM generated answers for creating thought graphs and final answers.

| Accuracy | GSM8K | AQUA | MBPP | ProofWriter |
|---|---|---|---|---|
| Answer for Creating Thought Graphs | 76.42 | 49.21 | 60.6 | 78.25 |
| Final Answers | 79.98 | 57.48 | 61.6 | 84.25 |
| Improvement | +3.56 | +8.27 | +1.00 | +6.00 |

### E.2 PERFORMANCE OF GRAPHIC WHEN USING INCORRECT ANSWERS FOR CREATING THOUGHT GRAPHS

To further investigate whether incorrect thought graphs could mislead the retrieval process, we selected a subset from each dataset, containing all queries associated with incorrect thought graphs. We evaluated GraphIC on these four subsets and compared its performance with that of top-performing baselines, both training-based (DQ-LoRe) and training-free (Complex-CoT). The results, shown in Table 6, reveal that even when GraphIC uses incorrect thought graphs to retrieve in-context examples, it still achieves a significant performance advantage. Note that the performance in the Table 6 is substantially lower than those in Table 1. This is because the queries that lead to incorrect thought graphs are typically the most difficult ones.

Table 6: Performance of GraphIC and top training-free/training-based baseline on the subset where GraphIC uses incorrect answers for creating thought graphs.

| Model | GSM8K | AQUA | MBPP | ProofWriter |
|---|---|---|---|---|
| Complex-CoT | 38.58 | 28.68 | 4.06 | 54.02 |
| DQ-LoRe | 40.83 | 33.33 | 16.75 | 65.51 |
| GraphIC | 43.08 | 33.33 | 15.23 | 70.11 |

### E.3 COMPARISON OF COMPUTATION TIME WITH OTHER SIMILAR BASELINES.

Our method belongs to the category of methods that use the generated output of LLMs to select in-context examples, which also includes methods such as Skill-kNN and DQ-LoRe. These approaches involve using the LLM during the retrieval phase, resulting in longer retrieval times compared to other baselines. However, by leveraging the power of LLMs, they are suitable for complex tasks. The computation times for the three models are presented in the Table 7. Specifically, the retrieval time for the GraphIC model is similar to that of DQ-LoRe, and slightly higher than Skill-kNN. Despite this, GraphIC significantly outperforms Skill-kNN in terms of performance. Moreover, compared to DQ-LoRe, which has the same retrieval time, GraphIC not only delivers superior performance but also greatly reduces both the prepare time and training time required by DQ-LoRe.

Here, "prepare time" refers to the time spent generating the necessary outputs for retrieval, such as generating the required skills for all candidate examples in Skill-kNN. For our evaluation, we used the GSM8K dataset with the LLM configured as Llama-3.1-8B-Instruct.

Table 7: Prepare time, training time, and retrieve time of GraphIC and other similar baselines.

| Time | Skill-kNN | DQ-LoRe | GraphIC |
|---|---|---|---|
| prepare time | 0.7h | 20h | 1.5h |
| traning time | - | 16h | - |
| retrieve time | 0.3s | 0.4s | 0.4s |

### E.4 PERFORMANCE OF GRAPHIC AND TOP TRAINING-BASED/TRAINING-FREE BASELINES ACROSS 1–8 SHOT SETTINGS

We find a difference in training-free methods compared to their training-based counterparts. Methods such as DQ-LoRe, which are training-based, directly optimize the probability of large language models (LLMs) producing correct answers in 1-shot scenarios. As a result, they tend to achieve superior performance in low-shot settings, particularly in 1-shot cases. However, as the number of shots increases, the performance gains of these methods may decelerate or even decline.

To further clarify this phenomenon, we conducted a comparison of GraphIC with the top-performing training-based and training-free baselines (DQ-LoRe and Complex-CoT) across 1–8 shot settings. The results, presented in Figure 6, highlight the strengths and weaknesses of training-based versus training-free approaches mentioned above.

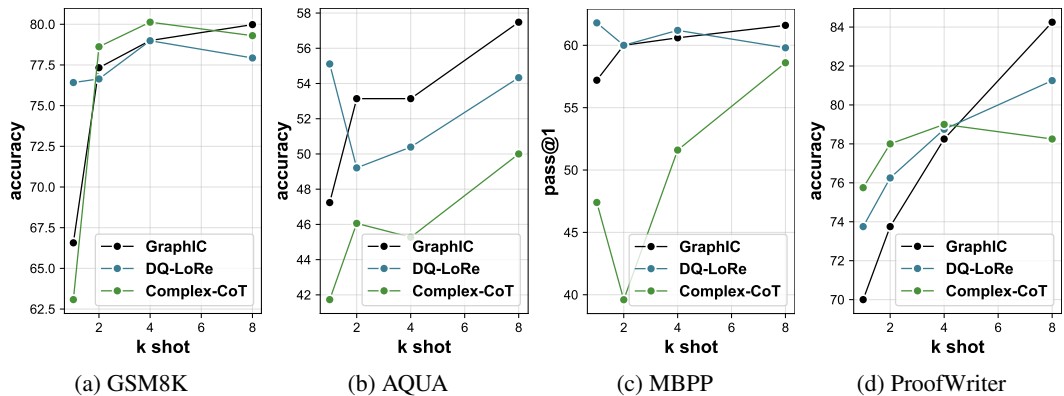

| (a) GSM8K | (b) AQUA | (c) MBPP | (d) ProofWriter |

Figure 6: Performance of GraphIC and Top Training-based/Training-free Baselines (DQ-LoRe and Complex-CoT) across 1–8 Shot Settings

## E.5    THE EFFECTS OF $\lambda$

We analyzed the effect of $\lambda$ values ranging from [0.0, 0.9] on the results across the four datasets utilized in our study. The corresponding results are presented in Figure 7, confirming the robustness of our method to the choice of lambda.

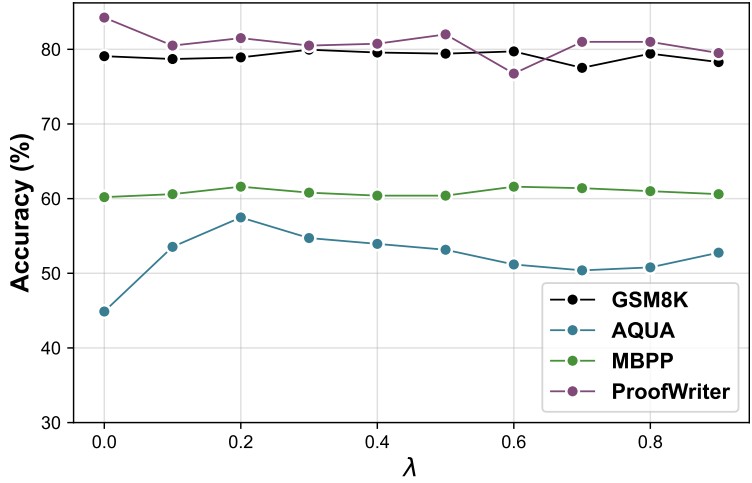

Figure 7: The performance of GraphIC on different $\lambda$ across four datasets.

## E.6    THE PERFORMANCE ON MATH DATASET

We conducted a comparison of GraphIC with the top-performing training-based and training-free baselines (DQ-LoRe and Complex-CoT) on MATH dataset. The results are presented in Table 8: the GraphIC model has consistently achieved optimal performance. Due to computational resource limitations, we randomly selected 500 samples from the training and testing data categorized as "level 5" difficulty in MATH, which were used as the candidate set and test set.

Table 8: The Performance of GraphIC, Complex-CoT, and DQ-LoRe on MATH Dataset

| Model | Complex-CoT | DQ-LoRe | GraphIC |
|-------|-------------|---------|---------|
| MATH  | 18.8        | 20.4    | 21.8    |

### E.7 THE PERFORMANCE ON MBPP WITH 16-SHOT SETTING

In order to analyze whether GraphIC can benefit from more than even more examples, we tested the performance of GraphIC and other top-3 baselines in the 16-shot setting on MBPP dataset. The experimental results are shown in the table below. The results indicate that while the performance of other baselines declines at 16 shots, GraphIC maintains an improvement. This makes GraphIC superior to the other baselines in the 16-shot scenario.

Table 9: The Performance of GraphIC and other top-3 baselines on MBPP with 16-shot Setting

| Model | BM25 | EPR | CEIL | GraphIC |
|---|---|---|---|---|
| MBPP | 60 | 60.6 | 60.6 | 62.6 |

## F SUPPLEMENTARY EXPLANATION

### F.1 THE MATRIX $B$

As shown in Figure 8, matrix $A$ represents the adjacency matrix of a thought graph, corresponding to the black edges in the graph. In this graph, vertices 1 and 2 have an in-degree of 0, indicating the starting points of reasoning, while vertices 3 and 4 have non-zero in-degrees, representing intermediate steps or results of reasoning. Matrix $B$ indicates the edges from vertices with non-zero in-degrees to those with in-degree 0, corresponding to the edges from vertices 3 and 4 to vertices 1 and 2 in the graph (marked in green). Since vertices 3 and 4 are intermediate steps or results of reasoning, and vertices 1 and 2 represent the starting points, these edges represent the retracing process.

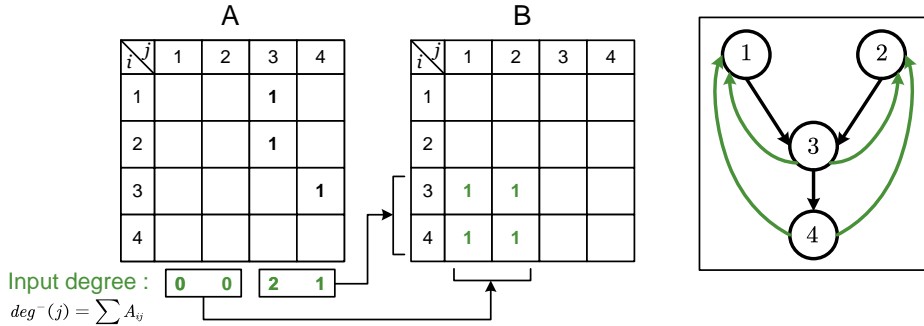

Figure 8: The definition of matrix $B$ and its corresponding edges in a graph.

### F.2 THE LOSS OF SETTING THE RANK OF MATRIX $W$ TO 1

First, we provide the optimal value and the optimal solution of the optimization problem defined by Equation 11, under the assumption that no constraints are imposed on $W$.

**Theorem F.1.** *Consider the following optimization problem:*

$$\max_{W} tr(ZW^\top X^\top), \quad W \in \mathbb{R}^{n_f \times n_f}, \quad s.t. \ ||W||_F = 1. \tag{17}$$

*The optimal value of this problem is $\left(\sum_{i=1}^{n_f} \sigma_i^2\right)^{1/2}$, and the optimal solution is $W^* = VYU^\top$, where $Y = diag(y_{11}, y_{22}, \ldots, y_{n_f n_f})$ with $y_{ii} = \frac{\sigma_i}{\left(\sum_{j=1}^{n_f} \sigma_j^2\right)^{1/2}}$.*

*Proof.* We begin by rewriting the objective function using the cyclic property of the trace:
$$\text{tr}(ZW^\top X^\top) = \text{tr}(W^\top X^\top Z).$$

Next, perform the Singular Value Decomposition (SVD) of $X^\top Z$:
$$X^\top Z = U\Sigma V^\top,$$
where $U$ and $V$ are orthogonal matrices, and $\Sigma = \text{diag}(\sigma_1, \sigma_2, \ldots, \sigma_{n_f})$ is the diagonal matrix of singular values.

Substituting this decomposition into the trace expression, we obtain:
$$\text{tr}(ZW^\top X^\top) = \text{tr}(W^\top U\Sigma V^\top).$$

Using the cyclic property of the trace again, we get:
$$\text{tr}(ZW^\top X^\top) = \text{tr}\left((V^\top W^\top U)\Sigma\right).$$

Define $Y = V^\top W^\top U$. Then the objective becomes:
$$\text{tr}(ZW^\top X^\top) = \text{tr}(Y\Sigma).$$

Since $U$ and $V$ are orthogonal, they preserve the Frobenius norm, i.e., $||Y||_F = ||W||_F = 1$. Thus, we are now tasked with maximizing $\text{tr}(Y\Sigma)$ subject to $||Y||_F = 1$.

We know that the trace $\text{tr}(Y\Sigma)$ is maximized when $Y$ is diagonal, i.e., when $Y = \text{diag}(y_{11}, y_{22}, \ldots, y_{n_f n_f})$. This follows because the off-diagonal elements of $Y$ do not contribute to $\text{tr}(Y\Sigma)$, but they affect the Frobenius norm of $Y$. By setting these off-diagonal elements to zero and redistributing the weight to the diagonal elements, we achieve a higher value of $\text{tr}(Y\Sigma)$.

Thus, the optimization problem reduces to:
$$\text{tr}(ZW^\top X^\top) = \sum_{i=1}^{n_f} \sigma_i y_{ii}, \quad \text{subject to} \quad \sum_{i=1}^{n_f} y_{ii}^2 = 1.$$

To find the optimal $y_{ii}$, we apply Cauchy-Schwarz inequality:
$$\sum_{i=1}^{n_f} \sigma_i y_{ii} \le \left(\sum_{i=1}^{n_f} \sigma_i^2\right)^{1/2} \left(\sum_{i=1}^{n_f} y_{ii}^2\right)^{1/2}.$$

Since $\sum_{i=1}^{n_f} y_{ii}^2 = 1$, this simplifies to:
$$\sum_{i=1}^{n_f} \sigma_i y_{ii} \le \left(\sum_{i=1}^{n_f} \sigma_i^2\right)^{1/2}.$$

The maximum value of $\sum_{i=1}^{n_f} \sigma_i y_{ii}$ is achieved when $y_{ii} = \frac{\sigma_i}{\left(\sum_{j=1}^{n_f} \sigma_j^2\right)^{1/2}}$. Thus, the optimal value of the objective is $\left(\sum_{i=1}^{n_f} \sigma_i^2\right)^{1/2}$, and the corresponding optimal solution is $W^* = VYU^\top$, where $Y = \text{diag}\left(\frac{\sigma_1}{\left(\sum_{i=1}^{n_f} \sigma_i^2\right)^{1/2}}, \ldots, \frac{\sigma_{n_f}}{\left(\sum_{i=1}^{n_f} \sigma_i^2\right)^{1/2}}\right)$. $\qquad\square$

Based on the proof above, we know that without any constraints on $W$, the optimal value of the optimization problem defined by Equation 11 is $\left(\sum_{i=1}^{n_f} \sigma_i^2\right)^{1/2}$. When a rank-1 constraint is added to $W$, the optimal value of the problem becomes $\sigma_1$. Therefore, we are interested in quantifying the loss introduced by the "rank-1 assumption," which can be assessed by the ratio of their optimal values, $r = \sigma_1 / \left(\sum_{i=1}^{n_f} \sigma_i^2\right)^{1/2}$. We computed the value of $r$ on four datasets, and the results are shown in Table 10. The table demonstrates that the loss caused by the "rank-1 assumption" is less than 0.1%, implying that it does not result in a significant loss of precision.

Table 10: The $r$ value on four datasets.

| Dataset | GSM8K | AQUA | MBPP | ProofWriter |
|---------|-------|------|------|-------------|
| $r$ | 0.99978 | 0.99991 | 0.99963 | 0.99921 |

### F.3 THE ASYMMETRY

We provide an example to illustrate the asymmetry. As shown in the example below, solving Question B includes solving Question A, which involves further calculations to determine how long it will take to reach the minimum age required by the company for employment. Therefore, in this case, referencing B can help resolve A, but referencing A does not necessarily resolve B.

```
Question A:
In 3 years, Jayden will be half of Ernesto's age. If Ernesto is 11 years old, how many years
    old is Jayden now?

Answer A:
Ernesto = 11 + 3 = 14
Jayden = 14/2 = 7 in 3 years
Now = 7 - 3 = 4
Jayden is 4 years old.

Question B:
The minimum age required to be employed at a company is 25 years. Dara aspires to work for the
    company and will be half the age of Jane in six years. If Jane is currently working for
    the company and is 28 years old, how long is it before Dara reaches the minimum age
    required by the company to be employed?

Answer B:
In six years, Jane will be 28+6 = 34 years old.
Dara will be half the age of Jane in six years, meaning she will be 34/2 = 17 years old in six
    years.
Currently, Dara is 17-6 = 11 years old.
Dara has to wait for 25-11 = 14 more years to reach the company's minimum age of employment.
```

### F.4 EXAMPLES OF THOUGHT GRAPHS ON MATH

**Candidate Example 1:**

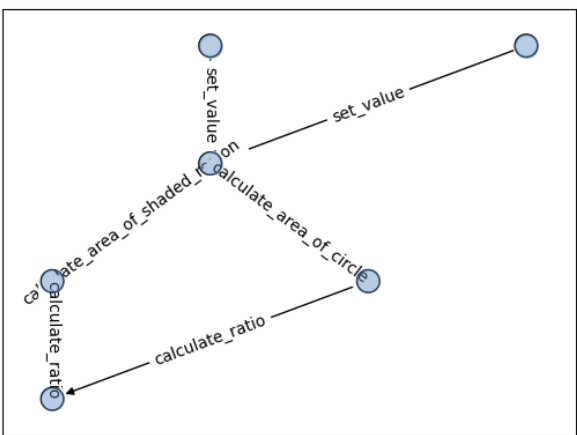

Figure 9: Thought graph of candidate example 1

```
Question:
[asy] fill(circle((4,0),4),grey); fill((0,0)--(8,0)--(8,-4)--(0,-4)--cycle,white); fill(circle
    ((7,0),1),white); fill(circle((3,0),3),white); draw((0,0)--(8,0),black+linewidth(1));
    draw((6,0)--(6,sqrt(12)),black+linewidth(1)); MP("A", (0,0), W); MP("B", (8,0), E); MP("C
    ", (6,0), S); MP("D",(6,sqrt(12)), N); [/asy]
In this diagram semi-circles are constructed on diameters $\overline{AB}$, $\overline{AC}$,
    and $\overline{CB}$, so that they are mutually tangent. If $\overline{CD} \bot \overline{
    AB}$, then the ratio of the shaded area to the area of a circle with $\overline{CD}$ as
    radius is:
$\textbf{(A)}\ 1:2\qquad \textbf{(B)}\ 1:3\qquad \textbf{(C)}\ \sqrt{3}:7\qquad \textbf{(D)}\
    1:4\qquad \textbf{(E)}\ \sqrt{2}:6$
```

```
Answer:
To make the problem much simpler while staying in the constraints of the problem, position
    point $C$ halfway between $A$ and $B$. Then, call $\overline{AC} = \overline{BC}=r$ . The
     area of the shaded region is then\[\frac{ \pi r^2 - \pi (r/2)^2 - \pi (r/2)^2}{2}=\frac
    {\pi r^2}{4}\]Because $\overline{CD}=r$ the area of the circle with $\overline{CD}$ as
    radius is $\pi r^2$. Our ratio is then\[\frac{\pi r^2}{4} : \pi r^2 = \boxed{1:4}\]
```

**Candidate Example 2:**

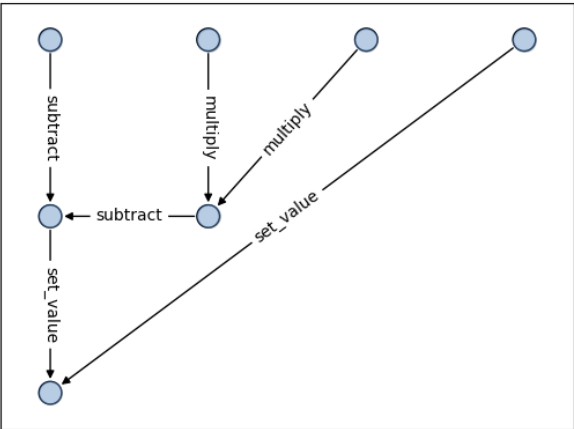

Figure 10: Thought graph of candidate example 2

```
Question:
The tail of a 1-mile long train exits a tunnel exactly 3 minutes after the front of the train
    entered the tunnel. If the train is moving 60 miles per hour, how many miles long is the
    tunnel?

Answer:
Because the train is moving 60 miles per hour, the front of the train moves 1 mile every
    minute. Therefore, in the three minutes since the front of the train entered the tunnel,
    the front of the train has moved three miles. At the end of these three minutes, we know
    the front of the train is 1 mile beyond the end of the tunnel,because the train is one
    mile long and its tail is just leaving the tunnel. So, the front of the train has moved 3
     miles from the beginning of the tunnel and is now 1 mile beyond the end of the tunnel.
    This tells us that the tunnel is $3-1 = \boxed{2\text{ miles}}$ long.
```

**Candidate Example 3:**

Figure 11: Thought graph of candidate example 3

```
Question:
On the refrigerator, MATHEMATICS is spelled out with $11$ magnets, one letter per magnet. Two
    vowels and four consonants fall off and are put away in a bag. If the T's, M's, and A's
    are indistinguishable, how many distinct possible collections of letters could be put in
    the bag?

Answer:
We count the number of ways to choose the vowels and the consonants separately. There are four
    vowels, of which two are As. If there are no As, then we must choose both the remaining
    vowels, so there is $1$ choice; if there is one A, then we can choose the remaining vowel
    in $2$ ways; and if there are two As, then there are no vowels left to choose, so there
    is $1$ choice. This makes $1 + 2 + 1 = 4$ distinct pairs of vowels.

There are seven consonants, of which two are Ts and of which two are Ms. Since we must choose
    four consonants, we must use at least one of the Ts and Ms.

If we use one T and no Ms, we have only $1$ choice (use the three remaining consonants); the
    same is true if we use one M and no Ts.
If we use both Ts and no Ms, there are $\tbinom{3}{2} = 3$ choices for the two remaining
    consonants; the same is true if we use both Ms and no Ts, or if we use one T and one M.
If we use both Ts and one M, there are $\tbinom{3}{1} = 3$ choices for the single remaining
    consonant; the same is true if we use both Ms and one T.
Finally, if we use both Ts and both Ms, there are no more letters left to choose, so we get
    $1$ more choice.

In total, we have $2(1) + 5(3) + 1 = 18$ distinct collections of consonants.

Therefore, the number of distinct collections of letters is $4 \cdot 18 = \boxed{72}.$
```

**Query 1:**

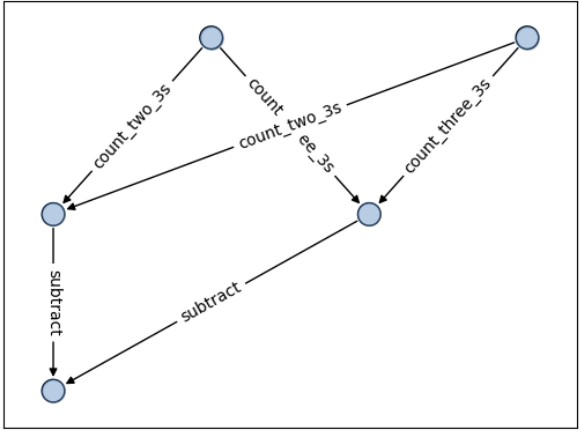

Figure 12: Thought graph of query 1

```
Question:
For each positive integer $n$, the set of integers $\{0,1,\ldots,n-1\}$ is known as the $\
    textit{residue system modulo}$ $n$. Within the residue system modulo $2^4$, let $A$ be
    the sum of all invertible integers modulo $2^4$ and let $B$ be the sum all of non-
    invertible integers modulo $2^4$. What is $A-B$?

Answer:
Since $2^4$ is a power of $2$, the invertible integers are the odd ones $
    \{1,3,5,7,9,11,13,15\}$, and the non-invertible integers are the even ones $
    \{0,2,4,6,8,10,12,14\}$. Thus, \begin{align*}
A-B & = (1+3+5+7+9+11+13+15)\\
& \qquad - (0+2+4+6+8+10+12+14)\\
& = (1-0)+(3-2)+(5-4)+(7-6)+(9-8)\\
&\qquad+(11-10)+(13-12)+(15-14)\\
& = 1+1+1+1+1+1+1+1=\boxed{8}.
\end{align*}
```

**Query 2:**

Figure 13: Thought graph of query 2

```
Question:
How many integers between 100 and 500 have at least two 3s as digits?

Answer:
First consider the two $3$s to appear in the units and tens places. Between $100$ and $500$,
    there are four such numbers: $133$, $233$, $333$, and $433$. Now consider the two $3$s to
     appear in the units and hundreds places. The numbers will be in the $300$s, so we don't
    need to worry about if they are between $100$ and $500$. There are $10$ choices for the
    tens digit, but we have already counted $333$, so such a scenario will add nine numbers.
    Finally, consider the two $3$s to appear in the tens and hundreds places. Again, these
    numbers are automatically between $100$ and $500$. There are $10$ choices for the units
    digit, but we again discard $333$ for a final count of nine such numbers. Thus, our
    answer is $4+9+9 = \boxed{22}$.
```

**Query 3:**

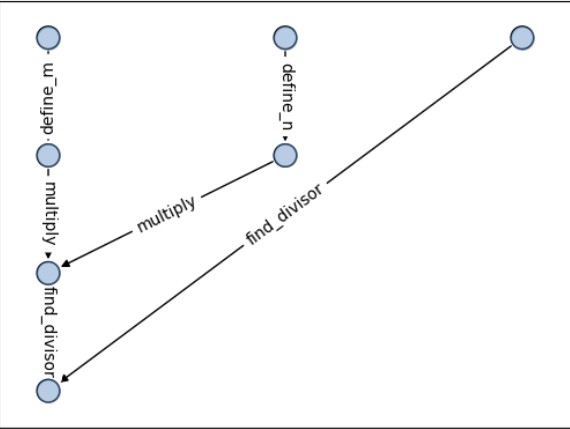

Figure 14: Thought graph of query 3

```
Question:
Given that $m$ and $n$ are positive integers such that $m\equiv 6\pmod 9$ and $n\equiv 0\pmod
    9$, what is the largest integer that $mn$ is necessarily divisible by?

Answer:
If $m\equiv 6\pmod 9$, then we can write $m$ as $9a+6$ for some integer $a$. This is equal to
    $3(3a+2)$, so $m$ is certainly divisible by $3$. If $n\equiv 0\pmod 9$, then $n$ is
    divisible by $9$. Therefore, $mn$ must be divisible by $3\cdot 9 = 27$.

Note that $m$ can be 6 and $n$ can be 9, which gives us $mn = 54$. Also, $m$ can be 15 and $n$
     can be 9, which gives us $mn = 135$. The gcd of 54 and 135 is 27.

Therefore, the largest integer that $mn$ must be divisible by is $\boxed{27}$.
```

