# OpenReview forum: "GraphIC: A Graph-Based In-Context Example Retrieval Model for Multi-Step Reasoning"
_ICLR.cc/2025/Conference — Submitted to ICLR 2025_

### Official Review · Reviewer_UqoT · 2024-10-28

**Soundness:** 3
**Presentation:** 2
**Contribution:** 3
**Rating:** 6
**Confidence:** 4

**Summary:**

This paper studied in-context example selection for ICL, and proposed a novel GraphIC method to capture reasoning processes for ICL in multi-step reasoning. GraphIC first generated thought graphs for candidate examples and query with LLM to encode the reasoning steps, and then employed a BN with PPR mechanism to studied the associated parameters reflecting the underlying reasoning structure. Finally, GraphIC selected the candidate examples with the parameter maximizing the probability of thought graph for the query in ICL. The authors conducted extensive experiments, and the results demonstrated that GraphIC outperformed both training-free and training-based baselines across various multi-step reasoning tasks.

**Strengths:**

1. The idea of selecting ICE with reasoning structure rather than semantics is novel and inspiring, and the human cognition-inspired methodology of capturing the underlying reasoning process with thought graph and BN is interesting.
2. The inference of the proposed method is solid, which makes the method more reasonable.
3. The authors conducted extensive experiments and deeper analysis to demonstrate the effectiveness of the proposed method and its components.

**Weaknesses:**

1. There are some concerns on the method design.
- According to the prompts in the Appendix, the proposed method seems to exploit the answer to generate the thought graph for both examples and the query. So where does the answer come from? If the answer is fetched from the annotations, it seems to lead to label leakage. If the answer is generated by the LLM, how to ensure its consistency to real solution?
- As the authors have mentioned in section 3, BN works on the DAG. However, there exist loops in thought graphs for codes which violate the DAG assumption as shown in Figure 3.
2. The presentation of the manuscript could be further improved especially for the confusing and unclear contents.
- The authors should explain the formal definition of the thought graph more clearly. Maybe one example could help. For example, what does one vertex and edge mean in the thought graph (does one vertex mean one step), how can it capture the reasoning processes, and how is the corresponding text attribute got. If the text is that in Figure 3, I do not think it contains much useful information. Besides, it would be better to explain the vertex, edge and random variable of BN in section 3 more clearly.
- Related works on BN should be cited properly.
- The insights behind the parameter W should be explained more clearly. Why the parameter can reflect the reasoning structure, and why the probability computed with parameter of the example and graph of the query can reflect their similarity on reasoning?
- The insights of the matrix B in PPR is confusing. What does the matrix mean and why can the matrix realize the retracing. Perhaps the authors could provide one example.
- The author mentioned that the asymmetry of the proposed method, so I wonder the advantages of the asymmetry compared with symmetry.
3. Some concerns on the experiments.
- The authors could add the LLM without ICL as one baseline to demonstrate the improvements more clearly.
- It would be better to add experiments to compare the computation time of the proposed method and other ICE selection baselines.
- In Figure 4, it seems that the proposed method does not work very well with 1-4 shot. It would be better to make clearer explanation.
- It would be better to add studies on the effects of lambda.

**Questions:**

See the Weaknesses.

---

> ### Author Response · Authors · 2024-11-20
> **Official Comment to Reviewer UqoT (1/4)**
>
> Thanks for the detailed comments and insightful suggestions! We will try our best to resolve the comments.
>
> **W1.1: According to the prompts in the Appendix, the proposed method seems to exploit the answer to generate the thought graph for both examples and the query. So where does the answer come from? If the answer is fetched from the annotations, it seems to lead to label leakage. If the answer is generated by the LLM, how to ensure its consistency to real solution?**
>
> Generating a thought graph indeed requires answers for both examples and the query. For candidate examples, we use annotated answers. For the query, we first generate an answer, then generate corresponding formalized reasoning representation, which is then parsed to construct the thought graph. There is no risk of label leakage in this process.
>
> Regarding your concerns about consistency, we cannot guarantee that the generated answer will always align perfectly with the real solution across all queries. However, we have the following reasons to explain why our method can achieve good performance.
>
> 1. Firstly, to analyze the consistency between LLM-generated answers and real solutions, we tested the accuracy of these answers used to generate the thought graphs. The results are shown in the table below. From this table, it can be seen that these answers used to generate the thought graph already have relatively high accuracy, which ensures their consistency with the real solution. Furthermore, the table demonstrates that using the thought graph to retrieve examples can further improve accuracy, especially in mathematical reasoning and logical reasoning tasks. This indirectly validates the consistency between the generated answers and the real solutions.
>
>    | Accuracy                              | GSM8K         | AQUA          | MBPP         | ProofWriter   |
>    | ------------------------------------- | ------------- | ------------- | ------------ | ------------- |
>    | answer (for generating thought graph) | 76.42         | 49.21         | 60.6         | 78.25         |
>    | final answer                          | 79.98 (+3.56) | 57.48 (+8.27) | 61.6 (+1.00) | 84.25 (+6.00) |
>
> 2. Secondly, even if the generated answers are not consistent with the real solution, it does not necessarily mean that the reasoning steps are entirely inconsistent; rather, many are "almost correct," involving only minor errors \[1]. This is also helpful for retrieving examples with similar reasoning processes. To further illustrate this fact, we selected a subset from each dataset, containing all queries associated with incorrect answers for generating thought graphs. We evaluated GraphIC on these four subsets and compared its performance with top training-based (DQ-LoRe) and training-free (Complex-CoT) baselines. The results, shown in the table below, reveal that even when GraphIC uses incorrect thought graphs to retrieve in-context examples, it still achieves a significant performance advantage. Note that the performance in the table below are substantially lower than those in Table 1. This is because the queries that lead to incorrect thought graphs are typically the most difficult ones.
>
>    | Model       | GSM8K     | AQUA      | MBPP      | ProofWriter |
>    | ----------- | --------- | --------- | --------- | ----------- |
>    | Complex-CoT | 38.58     | 28.68     | 4.06      | 54.02       |
>    | DQ-LoRe     | 40.83     | **33.33** | **16.75** | 65.51       |
>    | GraphIC     | **43.08** | **33.33** | 15.23     | **70.11**   |
>
> 3. Lastly, similar approaches have been applied in other ICE retrieval models and show good performance, such as Skill-kNN, which utilizes an LLM to generate the skills required to solve a problem, and DQ-LoRe, which uses an LLM to generate chain-of-thought reasoning answers.
>
> [1] Wei, Jason, et al. "Chain-of-thought prompting elicits reasoning in large language models." Advances in neural information processing systems 35 (2022): 24824-24837.

---

> ### Author Response · Authors · 2024-11-20
> **Official Comment to Reviewer UqoT (2/4)**
>
> **W1.2: As the authors have mentioned in section 3, BN works on the DAG. However, there exist loops in thought graphs for codes which violate the DAG assumption as shown in Figure 3.**
>
> First, we sincerely apologize for the confusion caused by referring to our probabilistic model as a "Bayesian Network" in the paper. We greatly value your feedback and will address this issue in the subsequent revised version. As you mentioned, classical BNs are inherently designed for DAGs. However, as stated in page 5 line 258, our approach draws inspiration from the core principles of BNs rather than directly implementing them, enabling our model to work on general directed graphs.
>
> In particular, we assume that in a directed graph, the value $x_i$ associated with each vertex $v_i$ is influenced by a hidden parameter $z_i$ and a matrix $W$. Formally, the probability distribution for each vertex is defined as $p(x_i; z_i, W) = \frac{1}{C_i} \exp\left[-(l - z_i^\top W x_i)\right]$, leading to a joint probability distribution for all vertices: $p(x_1, \dots, x_n; z_1, \dots, z_n, W) = \prod_{i=1}^n p(x_i; z_i, W)$. Notably, this formulation does not rely on the acyclic assumption. To estimate the hidden parameters $z_i$, we utilize an iterative procedure on the graph (as detailed in Equation 5). This process aggregates information from parent vertices to the current vertex, encapsulating the ideas of BN.
>
> **W2.1: The authors should explain the formal definition of the thought graph more clearly. Maybe one example could help. For example, what does one vertex and edge mean in the thought graph (does one vertex mean one step), how can it capture the reasoning processes, and how is the corresponding text attribute got. If the text is that in Figure 3, I do not think it contains much useful information. Besides, it would be better to explain the vertex, edge and random variable of BN in section 3 more clearly.**
>
> As depicted in the GSM8K subfigure of Figure 3, a thought graph is a directed graph where each vertex $v_i$ corresponds to a vector $x_i \in \mathbb{R}^{n_f}$. Each vertex's attribute $x_i$ is the embedding of a string (e.g., "add" or "multiply") and represents the corresponding operation. Edges in the graph represent the order of computations; for instance, an edge from the vertex labeled Emb("add") to the vertex labeled Emb("multiply") signifies performing addition first and then using the result in a multiplication. A similar conceptualization is also seen in works such as GraphMR [1]. This makes the thought graph an effective tool for capturing the sequence of operations that solve a mathematical problem, embodying core reasoning processes.
>
> The text attributes, such as "multiply" and "add," associated with each vertex are included in the formalized reasoning representation (FRR) (as shown in the brackets [] near the bottom of Figure 2) and are extracted during parsing. The corresponding text associated with each vertex is fundamental to our model. As previously explained, the embedding of this corresponding text represents the attribute $x_i$ of each vertex in the thought graph, enabling the calculation of the joint probability density of the vertices. We present the attributes of the vertices in the thought graph from Figure 3 in textual form in order to visually represent the proposed thought graph.
>
> As noted earlier, our probabilistic model is inspired by Bayesian Networks (BN) but is not a strict equivalent. In our model, $x_i$ represents the attribute values of vertices in the thought graph, while $z_i$ is computed through the iterative process that leverages the graph's connectivity information.
>
> Finally, we greatly appreciate your insightful suggestions. We will carefully revise our paper to provide a more precise explanation of the thought graph and our probabilistic model.
>
> [1] Feng, Weijie, et al. "GraphMR: Graph neural network for mathematical reasoning." Proceedings of the 2021 Conference on Empirical Methods in Natural Language Processing. 2021.
>
> **W2.2: Related works on BN should be cited properly.**
>
> We apologize for the inappropriate citation of work related to BN in our initial draft. We have since improved the relevant citations.

---

> ### Author Response · Authors · 2024-11-20
> **Official Comment to Reviewer UqoT (3/4)**
>
> **W2.3: The insights behind the parameter W should be explained more clearly. Why the parameter can reflect the reasoning structure, and why the probability computed with parameter of the example and graph of the query can reflect their similarity on reasoning?**
>
> In our probabilistic model, we assume that the closer $x_i$ is to $W z_i$, the more likely $x_i$ is to be generated. Here, $x_i$ represents the operation of a specific vertex, and $z_i$, calculated through the iterative process described above, mainly incorporates information from the parent vertices of $x_i$. This indicates that $W$ determines the next operation based on the preceding ones. For instance, in a thought graph with two vertices labeled ${x_1 = \text{Emb("add")}, x_2 = \text{Emb("multiply")}}$ and an edge $(v_1, v_2)$, $W$ learns a mapping from $x_1$ to $x_2$, representing that “multiply” follows “add.”
>
> When this $W$ is used as a parameter for another thought graph, a high joint probability implies that the graph also contains frequent “multiply” → “add” operations, suggesting that the problem-solving processes of the two graphs are similar. The intuition underlying $W$ reflects a comparable concept: If the parameters estimated from one set of data points perform well when applied to another set of data points, it indicates that the two sets of data points are very similar.
>
> **W2.4: The insights of the matrix B in PPR is confusing. What does the matrix mean and why can the matrix realize the retracing. Perhaps the authors could provide one example.**
>
> Firstly, we have added a diagram in the appendix F.1 (Figure 8) to better illustrate the role of matrix B. Matrix $B \in \mathbb{R}^{n \times n}$ is an adjacency matrix, and its corresponding graph is defined as follows: for any pair of vertices $i$ and $j$, a directed edge from $i$ to $j$ exists if vertex $i$ has a degree greater than 0 and $j$ has a degree of 0. A vertex with a degree of 0 indicates that it is not derived from other operations and thus serves as the starting point of reasoning. Consequently, matrix $B$ represents the edges connecting all other vertices to the reasoning starting points, illustrating the retracing process during reasoning.
>
> **W2.5: The author mentioned that the asymmetry of the proposed method, so I wonder the advantages of the asymmetry compared with symmetry.**
>
> First, asymmetry is a fundamental characteristic of real-world situations. For example, as the example shown in Appendix F.3, mathematical problem A might be a subproblem of problem B. In such a case, referencing B may resolve A, but referencing A does not necessarily resolve B. Second, our experiments show that actual scenario is asymmetry. As shown in Figure 5a, the value at the $i$-th row and $j$-th column corresponds to the probability of the LLM correctly predicting the answer of example $j$ when example $i$ is used as an in-context example. The heatmap in Figure 5a is evidently asymmetric, as the value at (i, j) does not equal the value at (j, i).
>
> **W3.1: The authors could add the LLM without ICL as one baseline to demonstrate the improvements more clearly.**
>
> We sincerely appreciate your insightful suggestion. In response, we tested the baseline model without ICL for Llama-3.1-8B-Instruct, and the results are shown below. The experimental results show that ICL significantly enhances multi-step reasoning performance.
>
> | Dataset | GSM8K | AQUA  | MBPP | ProofWriter |
> | ------- | ----- | ----- | ---- | ----------- |
> | w/o ICL | 46.47 | 35.43 | 43.4 | 40.75       |

---

> ### Author Response · Authors · 2024-11-20
> **Official Comment to Reviewer UqoT (4/4)**
>
> **W3.2: It would be better to add experiments to compare the computation time of the proposed method and other ICE selection baselines.**
>
> Our method belongs to the category of methods that use the generated output of LLMs to select in-context examples, which also includes methods such as Skill-kNN and DQ-LoRe. These approaches involve using the LLM during the retrieval phase, resulting in longer retrieval times compared to other baselines. However, by leveraging the power of LLMs, they are suitable for complex tasks. The computation times for the three models are presented in the table below. Specifically, the retrieval time for the GraphIC model is similar to that of DQ-LoRe, and slightly higher than Skill-kNN. Despite this, GraphIC significantly outperforms Skill-kNN in terms of performance. Moreover, compared to DQ-LoRe, which has the same retrieval time, GraphIC not only delivers superior performance but also greatly reduces both the prepare time and training time required by DQ-LoRe.
>
> |               | Skill-kNN | DQ-LoRe | GraphIC |
> | ------------- | --------- | ------- | ------- |
> | prepare time  | 37min     | 20h     | 1.5h    |
> | traning time  | -         | 16h     | -       |
> | retrieve time | 0.30s     | 0.4s    | 0.4s    |
>
> Here, "prepare time" refers to the time spent generating the necessary outputs for retrieval, such as generating the required skills for all candidate examples in Skill-kNN. For our evaluation, we used the GSM8K dataset with the LLM configured as Llama-3.1-8B-Instruct.
>
> **W3.3: In Figure 4, it seems that the proposed method does not work very well with 1-4 shot. It would be better to make clearer explanation.**
>
> Our method does not work very well with 1-4 shot, where it generally underperforms relative to training-based baselines. This reflects a difference of training-free methods compared to their training-based counterparts. Methods such as DQ-LoRe, which are training-based, directly optimize the probability of large language models (LLMs) producing correct answers in 1-shot scenarios. As a result, they tend to achieve superior performance in low-shot settings, particularly in 1-shot cases. However, as the number of shots increases, the performance gains of these methods may decelerate or even decline.
>
> To further clarify this phenomenon, we conducted a comparison of GraphIC with the top-performing training-based and training-free baselines (DQ-LoRe and Complex-CoT) across 1–8 shot settings. The results, presented in the Appendix E.4 (Figure 6), highlight the strengths and weaknesses of training-based versus training-free approaches metioned above.
>
> Furthermore, 8-shot is the most commonly used setting in chain-of-thought scenarios \[1]\[2]\[3], so we mainly focus on the performance in the 8-shot setting.
>
> [1] Wei, Jason, et al. "Chain-of-thought prompting elicits reasoning in large language models." Advances in neural information processing systems 35 (2022): 24824-24837.
>
> [2] Zhang, Zhuosheng, et al. "Automatic Chain of Thought Prompting in Large Language Models." The Eleventh International Conference on Learning Representations.
>
> [3] Xiong, Jing, et al. "DQ-LoRe: Dual Queries with Low Rank Approximation Re-ranking for In-Context Learning." The Twelfth International Conference on Learning Representations.
>
> **W3.4: It would be better to add studies on the effects of lambda.**
>
> In response, we analyzed the effect of lambda values ranging from [0.0, 0.9] on the results across the four datasets utilized in our study. The corresponding results are presented in the appendix E.5 (Figure 7), confirming the robustness of our method to the choice of lambda.

---

> > ### Comment · Reviewer_UqoT · 2024-11-22
> >
> > Thanks for your responses. It has addressed most of my concerns. I still have some concerns regarding your responses.
> >
> > For 1.1, the proposed method assumes that the generated answers agree with the true answers to some extent, so could the method generalize to more complex datasets, such as MATH or AIME that the LLM may generate totally wrong answers.
> >
> > For 1.2, it seems that your implementation is the same as the preliminaries, so what's your improvement.
> >
> > For 2.1, I'm still confusing what the node and edge in the thought graph mean. Does one node mean one reasoning step, or it just has totally different meanings in different datasets and has no common ideas. Besides, I still wonder whether the text attribute could provide enought information. For example, compared with the "multiply" in the solution, we may care more about the insights behind "why multiply", which is absent in the graph.
> >
> > For 3.2, the authors may need to explain the three times more clearly. The proposed method includes generating thought graphs for both candidates and query, estimating the parameters for each candidate, and computing the probability for the query with each candidate parameters, it should be make clearer each step belongs to which group.

---

> > > ### Author Response · Authors · 2024-11-22
> > > **Official Comment to Reviewer UqoT (2/2)**
> > >
> > > > For 2.1, I'm still confusing what the node and edge in the thought graph mean. Does one node mean one reasoning step, or it just has totally different meanings in different datasets and has no common ideas. Besides, I still wonder whether the text attribute could provide enough information. For example, compared with the "multiply" in the solution, we may care more about the insights behind "why multiply", which is absent in the graph.
> > >
> > > **The meaning of nodes and edges in thought graph:**
> > >
> > > First, the nodes and edges in a thought graph share a common idea. A node represents a step in the reasoning process, while an edge captures the dependency between steps (i.e., a child step can only proceed after the parent step is completed). However, due to the intrinsic differences across tasks, the definition of node labels differs between tasks.
> > >
> > > For example, in mathematical reasoning tasks (such as GSM8K, AQUA, MATH), a reasoning step can be expressed as "which mathematical operations were performed on which variables"; in code generation tasks (such as MBPP), it can be expressed as "what code was executed"; and in logical reasoning tasks (such as ProofWriter), it can be expressed as "which existing conclusions were used to derive new conclusions". Therefore, in these three distinct tasks, node labels are defined as descriptions of mathematical operations, code snippets, and intermediate conclusions, respectively.
> > >
> > > **Concern about text attributes:**
> > >
> > > Regarding your concern about whether using only operations as text attributes can provide enough information, we argue that in our scenario, this approach is sufficient for retrieving similar examples. While considering a single operation (such as "multiply") in isolation, its underlying meaning can be diverse. However, when the entire thought graph is given, the insight behind each operation is relatively clear. For instance, if a "divide" operation follows a "minus" operation, it is most likely used to calculate the rate of change of one quantity relative to another. This is similar to the following fact: even though a word may have different meanings, its meaning is clear when the context is provided. Thus, when the entire thought graph is similar, the insights behind the operations tend to be similar as well, even if the insight behind each operation is not explicitly expressed.
> > >
> > > Furthermore, we believe explicitly representing the insights behind each operation could enhance the performance and interpretability of in-context example retrieval. However, this information is often obscured by a large amount of shallow semantics. If we directly use text or text embedding to represent the insight behind each step, shallow semantics such as scene information, object names will inevitably be introduced, which may mislead the retrieval process. How to decouple the insight behind each step from the original text information is a topic of great research value and one that we are curious about, which requires deeper investigation.
> > >
> > > > For 3.2, the authors may need to explain the three times more clearly. The proposed method includes generating thought graphs for both candidates and query, estimating the parameters for each candidate, and computing the probability for the query with each candidate parameters, it should be make clearer each step belongs to which group.
> > >
> > > The computation process is divided into three stages: prepare time, training time, and retrieve time, described as follows:
> > >
> > > 1. **Prepare time** involves the time required to use LLMs to generate necessary outputs and complete other preparations for training and retrieval. For instance, Skill-kNN uses LLMs to generate the skills required for each candidate and computes their embedding. Similarly, DQ-LoRe uses extensive LLM calls to generate chain-of-thought answers required to train the retrieval model.
> > > 2. **Training time** involves the time required to train a retrieval model. For example, DQ-LoRe involves training a dense retrieval model.
> > > 3. **Retrieve time** involves the time required to retrieve k examples for a given query. For example, Skill-kNN generates the corresponding skill for query and uses cosine similarities between query skill embedding and candidate skill embedding for retrieval. DQ-LoRe generates a chain-of-thought answer for the query and employs the dense retrieval model to retrieve in-context examples.
> > >
> > > Each step in GraphIC and its corresponding group are shown in the table below.
> > >
> > > | Group         | Step                                                         |
> > > | ------------- | ------------------------------------------------------------ |
> > > | Prepare time  | generating thought graphs for candidates, estimating the parameters for each candidate |
> > > | Training time | -                                                            |
> > > | Retrieve time | generating thought graphs for query, computing the probability for the query with each candidate parameters |

---

> > > > ### Comment · Reviewer_UqoT · 2024-11-26
> > > >
> > > > Thanks a lot for your responses. The responses have addressed my major concerns, which I think should be added to make the paper clearer. I have raised my scores.

---

> > > > > ### Author Response · Authors · 2024-11-26
> > > > > **Official Comment to Reviewer UqoT**
> > > > >
> > > > > We sincerely appreciate the time and effort you invested in carefully reading our paper. Your detailed and valuable suggestions have been immensely helpful. We will incorporate the relevant content into the revised paper to enhance its clarity. If you still have any concerns or questions, please feel free to let us know, and we will do our best to address them promptly.

---

> ### Author Response · Authors · 2024-11-22
> **Official Comment to Reviewer UqoT (1/2)**
>
> We sincerely appreciate your prompt and thoughtful response to our previous comments. We will try our best to resolve your concerns.
>
> > For 1.1, the proposed method assumes that the generated answers agree with the true answers to some extent, so could the method generalize to more complex datasets, such as MATH or AIME that the LLM may generate totally wrong answers.
>
> In response, we first compared GraphIC with the top-performing training-based and training-free baselines (DQ-LoRe and Complex-CoT) on MATH dataset. The results are shown in the table below. The results show that GraphIC has consistently achieved optimal performance. Due to the time limit during the rebuttal period, we randomly selected 500 samples from the training and testing data categorized as "level 5" difficulty in MATH, which were used as the candidate set and test set.
>
> |      | Complex-CoT | DQ-LoRe | GraphIC  |
> | ---- | ----------- | ------- | -------- |
> | MATH | 18.8        | 20.4    | **21.8** |
>
> Secondly, we evaluated the accuracy of the answers for generating thought graphs. The results indicate that the accuracy of these answers is only 19%, meaning that most of the thought graphs are incorrect. However, GraphIC still achieves the best performance, demonstrating high robustness to the accuracy of thought graphs.
>
> Finally, we tested the performance of GraphIC and other top-training-based and training-free baselines on the subset containing all queries whose thought graphs are generated using incorrect answers. The results are shown below. GraphIC also achieved the best performance in this subset, further confirming its robustness to the accuracy of thought graphs.
>
> |          | Complex-CoT | DQ-LoRe | GraphIC   |
> | -------- | ----------- | ------- | --------- |
> | MATH (subset) | 12.34       | 13.82   | **14.07** |
>
> > For 1.2, it seems that your implementation is the same as the preliminaries, so what's your improvement.
>
> Compared to to the classical BNs in the preliminaries, we have made the following improvements:
>
> 1. In classical BNs, the hidden parameter $z_i$ for vertex $i$ is derived solely from its parent vertices. In contrast, our approach computes $z_i$ using the information of more nodes such as more predecessors of node $i$, which is achieved through an iterative process on the graph (as shown in Equation 4). This change is motivated by the following considerations:
>    - As noted on page 5, line 258, this iterative approach better aligns with human reasoning, which often involves iteratively reviewing information from earlier steps or returning to the beginning to re-examine the entire reasoning process.
>    - As discussed in response 1.2, our probabilistic model first performs the iterative process on the graph and then calculates the probability for each vertex. This does not require assuming that the thought graph is a DAG, which enabling us to extend our approach to more tasks such as code generation.
> 2. For BNs in continuous scenario, $p(x_i | \text{pa}(v_i))$ is typically modeled as a Gaussian distribution, which can be viewed as the squared Mahalanobis distance employed in Preliminaries Equation 2 for $\text{dist}(\cdot)$. We replace this with the distance function defined in Equation 6, derived from the inner product of vectors, which is more suitable for our scenario.

---

### Official Review · Reviewer_eg3J · 2024-10-29

**Soundness:** 3
**Presentation:** 3
**Contribution:** 3
**Rating:** 5
**Confidence:** 3

**Summary:**

This paper studies how to retrieve demonstration examples more effectively in in-context learning to improve the performance of in-context learning. Specifically, the paper proposes a method named GraphIC, which establishes thought graphs and models them as Bayesian networks, then retrieves demonstration examples that make the probability density of queries' bayesian networks maximum. Experiments show the effectiveness of the proposed method.

**Strengths:**

* The motive of the paper is reasonable and the method proposed is novel.
* Writing of this paper is good, with reasonable structure.
* The experiments are relatively abundant, and the experimental results can prove the conclusion of the paper.

**Weaknesses:**

* Some parts of the method section of the paper lack some details, there are many assumptions but no conditions, refer to questions.
* Method relies on LLM to construct a thought graph, which may be difficult or inaccurate to decompose key steps for complex problems.
* The lack of experiments on the thought graph, in my opinion, is an important part of the method and has a big impact on method performance, refer to questions.

**Questions:**

* In the 'Estimation of Model Parameters' paragraph, the rank of W is set to 1, there is no theoretical basis for this hypothesis, and how much precision loss is caused?
* As mentioned in weakness, LLM is probably hard to construct thought graph, do you try more complex datasets, such as widely used mathematical MATH dataset [1]?
* How accurate is the LLM at generating a thought graph?
* Multiple generations of the LLM may produce different thought graphs, how much will this affect the results?

[1] https://github.com/hendrycks/math/

---

> ### Author Response · Authors · 2024-11-20
> **Official Comment to Reviewer eg3J**
>
> Thanks for the detailed comments and insightful suggestions! We will try our best to resolve the comments.
>
> **Q1: In the 'Estimation of Model Parameters' paragraph, the rank of W is set to 1, there is no theoretical basis for this hypothesis, and how much precision loss is caused?**
>
> Through both theoretical and experimental analysis, we demonstrate that setting the rank of $W$ to 1 incurs almost no loss. First, we theoretically prove that if no assumptions are made about the matrix $W$, the optimal value of the optimization problem defined by Equation 11 is $\left( \sum_{i=1}^{n_f} \sigma_i^2 \right)^{1/2}$, as shown in Appendix F.2. If the rank of the matrix is set to 1, the optimal value is $\sigma_1$. Here, $\sigma_1, \dots, \sigma_{n_f}$ represent the singular values of matrix $W$.
>
> When $W$ is a low-rank matrix, $\sigma_1$ (the largest singular value) is much greater than the other $\sigma_i$. This causes $\sigma_1$ and $\left( \sum_{i=1}^{n_f} \sigma_i^2 \right)^{1/2}$ to be approximately equal, which implies that setting the rank of matrix $W$ to 1 is approximately lossless. We computed the ratio between these two optimal values, $\sigma_1 / \left( \sum_{i=1}^{n_f} \sigma_i^2 \right)^{1/2}$, on the four datasets used to evaluate the loss caused by the "rank-1 assumption". The experimental results show that the optimal values differ by less than 0.1%, which makes the process approximately lossless.
>
> | Dataset | GSM8K   | AQUA    | MBPP    | ProofWriter |
> | ------- | ------- | ------- | ------- | ----------- |
> | ratio   | 0.99978 | 0.99991 | 0.99963 | 0.99921     |
>
> **Q2: As mentioned in weakness, LLM is probably hard to construct thought graph, do you try more complex datasets, such as widely used mathematical MATH dataset [1]?**
>
> We sincerely appreciate your insightful suggestion. In response, we conducted a comparison of GraphIC with the top-performing training-based and training-free baselines (DQ-LoRe and Complex-CoT) on MATH dataset. The results are presented as follows: the GraphIC model has consistently achieved optimal performance. Due to computational resource limitations, we randomly selected 500 samples from the training and testing data categorized as "level 5" difficulty in MATH, which were used as the candidate set and test set.
>
> |      | Complex-CoT | DQ-LoRe | GraphIC  |
> | ---- | ----------- | ------- | -------- |
> | MATH | 18.8        | 20.4    | **21.8** |
>
> **Q3: How accurate is the LLM at generating a thought graph?**
>
> We cannot guarantee that the thought graph generated by the LLM is entirely accurate. However, our experimental results show that the thought graph generated by the LLM is sufficiently accurate. This is because relying on the generated thought graph enables optimal performance across all datasets. Furthermore, when an LLM fails to decompose the key steps of complex problems to construct a thought graph, it does not mean that the thought graph and the ground truth are entirely different. In fact, they share many similarities, with only a few steps differing. This also proves helpful in the retrieval process of in-context learning.
>
> **Q4: Multiple generations of the LLM may produce different thought graphs, how much will this affect the results?**
>
> We also considered that the LLM's multiple outputs could lead to different results. As described in the implementation details, the temperature of the LLM was set to 1e-5 for the generation of the thought graph. This setting ensures method stability, enabling identical results across multiple executions.

---

> > ### Comment · Reviewer_eg3J · 2024-11-24
> > **Thanks for your comments**
> >
> > Thank you very much for your detailed comments. However, I still have a few questions that require further clarification:
> >
> > 1. **Regarding Q2 and Q3**: Could you provide specific examples from the MATH dataset to illustrate whether the model can generate meaningful thought graphs? If the model is unable to generate such graphs, from where do the performance improvements of GraphIC originate?
> >
> > 2. **Regarding Q4**: What I would like to know is GraphIC's robustness w.r.t. diverse LLM generations, which would also help reduce the effects of LLM hallucinations.

---

> > > ### Author Response · Authors · 2024-11-25
> > > **Official Comment to Reviewer eg3J**
> > >
> > > We sincerely appreciate your feedback and will make every effort to address any concerns you might still have.
> > >
> > > > **Regarding Q2 and Q3**: Could you provide specific examples from the MATH dataset to illustrate whether the model can generate meaningful thought graphs? If the model is unable to generate such graphs, from where do the performance improvements of GraphIC originate?
> > >
> > > In response, we selected six thought graphs from the MATH dataset—three from the candidate set and three from the test set—and included them, along with their corresponding questions and answers, in Appendix F.4. The examples provided illustrate that LLMs can generate meaningful thought graphs, thereby improving the retrieval of in-context examples. Furthermore, if you want to explore more thought graphs, we have included all the thought graphs of the MATH dataset in the supplementary materials (in math.zip).
> > >
> > > > **Regarding Q4**: What I would like to know is GraphIC's robustness w.r.t. diverse LLM generations, which would also help reduce the effects of LLM hallucinations.
> > >
> > > We sincerely apologize for previously misunderstanding your concern. In response, we set the temperature of the LLM to 1 and conducted five experiments on GSM8K to evaluate how diverse generations influence the performance of GraphIC. The results are summarized in the table below. This result demonstrates the robustness of GraphIC to the diverse generation of LLMs. For our experiments, we utilized Llama-3.1-8B-instruct as LLM.
> > >
> > > |       | 1     | 2     | 3     | 4     | 5     | Mean (Std. Dev.) |
> > > | ----- | ----- | ----- | ----- | ----- | ----- | ---------------- |
> > > | GSM8K | 79.07 | 80.59 | 79.23 | 79.91 | 79.68 | 79.70 (0.54)     |
> > >
> > > Moreover, we also confirmed that GraphIC is robust to variations in LLM selection. As evidenced in Table 1 and Table 2, GraphIC consistently achieved top-tier performance across three distinct LLMs: GPT-4o-mini, Llama-3.1-8B-Instruct, and GPT-3.5-Turbo.

---

> > > ### Author Response · Authors · 2024-12-02
> > > **Official Comment to Reviewer eg3J**
> > >
> > > Dear Reviewer eg3J,
> > >
> > > We would like to sincerely thank you for your time and valuable feedback. Could you kindly confirm if our responses have adequately addressed your concerns? If there are any remaining issues, we would greatly appreciate it if you could point them out so we can make further improvements. If the concerns have been fully addressed, we would be grateful if you could consider updating your score accordingly. Thank you again for your thoughtful consideration.
> > >
> > > Best regards,
> > >
> > > Authors

---

> > > > ### Comment · Reviewer_eg3J · 2024-12-03
> > > >
> > > > Thanks for your response. To some extent, it solves my concerns. CurrentlyI feel like maintaining my score because I still doubt about what the thought graph actually means and how it performs in hard problems.

---

> > > > > ### Author Response · Authors · 2024-12-03
> > > > > **Official Comment to Reviewer eg3J**
> > > > >
> > > > > We sincerely appreciate your feedback and will make every effort to address any concerns you might still have.
> > > > >
> > > > > **What the thought graph actually means:**
> > > > >
> > > > > A thought graph is a graph in which vertices are assigned attributes, and it serves to represent the reasoning process. In a thought graph, a node represents a step in the reasoning process, while an edge captures the dependency between steps (i.e., a child step can only proceed after the parent step is completed).
> > > > >
> > > > > For example, as depicted in the GSM8K subfigure of Figure 3, a thought graph is a directed graph where each vertex $v_i$ corresponds to a vector $x_i \in \mathbb{R}^{n_f}$. Each vertex's attribute $x_i$ is the embedding of a string (e.g., "add" or "multiply") and represents the corresponding operation. Edges in the graph represent the order of computations; for instance, an edge from the vertex labeled Emb("add") to the vertex labeled Emb("multiply") signifies performing addition first and then using the result in a multiplication. A similar conceptualization is also seen in works such as GraphMR [1]. This makes the thought graph an effective tool for capturing the sequence of operations that solve a mathematical problem, embodying core reasoning processes.
> > > > >
> > > > > Moreover, thought graph is also explained in our responses to W2.1 of Reviewer UqoT and Q1 of Reviewer 1SZm.
> > > > >
> > > > > [1] Feng, Weijie, et al. "GraphMR: Graph neural network for mathematical reasoning." Proceedings of the 2021 Conference on Empirical Methods in Natural Language Processing. 2021.
> > > > >
> > > > > **How it performs in hard problems:**
> > > > >
> > > > > The MATH dataset is widely acknowledged as a hard task that involves multiple reasoning steps. Our experiments on the MATH dataset demonstrate that, even for the hard tasks, GraphIC continues to outperform other methods. The relevant results can be found in our reply to Q2.
> > > > >
> > > > > Additionally, we have included the thought graphs corresponding to the MATH dataset in Appendix F.4 and the attached file math.zip. Please kindly check them at your convenience.

---

> > > > > ### Author Response · Authors · 2024-12-04
> > > > > **Official Comment to Reviewer eg3J**
> > > > >
> > > > > Dear Reviewer eg3J,
> > > > >
> > > > > We would like to sincerely thank you for your time and valuable feedback. Could you kindly confirm if our responses have adequately addressed your concerns? If the concerns have been addressed, we would be grateful if you could consider updating your score accordingly. Thank you again for your thoughtful consideration.
> > > > >
> > > > > Best regards,
> > > > >
> > > > > Authors

---

> ### Author Response · Authors · 2024-12-01
> **Official Comment to Reviewer eg3J**
>
> We would like to sincerely thank you for your time and efforts. We kindly ask whether our responses have addressed your concerns. If there are any concerns that have not been fully addressed, please let us know. We will make every effort to resolve them.

---

### Official Review · Reviewer_1SZm · 2024-11-03

**Soundness:** 2
**Presentation:** 1
**Contribution:** 3
**Rating:** 5
**Confidence:** 3

**Summary:**

This paper proposes GraphIC, a graph-based method for in-context example retrieval aimed to multi-step reasoning tasks. GraphIC models CoTs as "tought graphs, and uses Bayesian networks and personalised pagerank for selecting in-context examples with similar underlying graph structures. Empirical results show marginal improvements or competitive results on a wide array of reasoning tasks (GSM8K, AQUA, MBPP, ProofWriter).

Bayesian Networks (BNs) are a common way to represent complex joint distributions over sets of variables; however, this work never formalises which joint distribution the BNs in this work aim to represent. Furthermore, in Eq. 2, it is not really true that "commonly, in BNs, $p(x_i | pa(v_i)) = g(dist(x_i, \hat{x}_i))$" -- BNs aim at representing dependence distributions between variables, like "rainy weather" and "risk of aqua planning", and it's not really clear what a distance function between those variables should represent.

Furthermore, the paper is based on the construction of "thought graphs", but it's not really clear what these are -- are they task dependent? How do they look like? Given a taks, how does someone create a "thought graph"? The paper also says that "to facilitate computation, we further represent the vertex attribures as the BERT embedding of corresponding text" -- what is going on exactly?

Then, my understanding is that the BN defines a distribution over "thoguht graphs", and that the distribution over $\hat{x}_i$; what does $\hat{x}_i$ represent? (E.g. in Eq. 8)

Results seem to be marginally better than the baselines in most cases, but it's really hard to understand what's going on in the proposed method.

**Strengths:**

The proposed method seems to produce marginally better results than the baselines in most cases.

**Weaknesses:**

It is very hard to understand what's going on in the proposed method -- for example, the method uses Bayesian networks but the paper never explicitly states which join distribution the Bayesian network aims to represent.

**Questions:**

Can you please explicitly state, e.g., what joint the BN is representing? What exactly is a "graph of thought"? Can you please provide a brief, clear, and self-contained description of the proposed method?

---

> ### Author Response · Authors · 2024-11-20
> **Official Comment to Reviewer 1SZm (1/2)**
>
> We apologize for the confusion caused and will make every effort to resolve your concerns.
>
> > Bayesian Networks (BNs) are a common way to represent complex joint distributions over sets of variables; however, this work never formalises which joint distribution the BNs in this work aim to represent.
>
> We apologize for any confusion caused by the insufficient explanation of the joint distribution we aim to model. The intended joint probability distribution is as follows: consider a multi-step reasoning task where the sequence of actions is denoted as $x_1, x_2, \dots, x_n$. Our objective is to model the joint probability density of $x_1, x_2, \dots, x_n$ and estimate the probability of correctly solving a multi-step reasoning problem by calculating the likelihood of executing the correct sequence of actions. For example, to solve a mathematical problem involving three steps, we aim to calculate the probability of a problem solver sequentially taking steps under the given distribution parameters ("add a and b to get c", "subtract d from c to get e", "multiply a and e to get f").
>
> Furthermore, we sincerely apologize for the confusion caused by referring to our probabilistic model as a "Bayesian Network" in the paper. It should be noted that while our approach is inspired by BNs, it has some differences. Specifically, to model this sequence, we assume that $x_1, x_2, \dots, x_n$ correspond to the vertices $v_1, v_2, \dots, v_n$ of a directed graph. In this graph, the value of $x_i$ is influenced by a hidden parameter $z_i$ and a matrix $W$. Formally, the probability distribution for each vertex is given by $p(x_i; z_i, W) = \frac{1}{C_i} \exp\left[-(l - z_i^\top W x)\right]$, resulting in the joint probability distribution for all vertices: $p(x_1, \dots, x_n; z_1, \dots, z_n, W) = \prod_{i=1}^n p(x_i; z_i, W)$. To estimate the hidden parameters \(z_i\), we adopt an iterative procedure on the graph (described in Equation 5), which aggregates information from parent vertices to the current vertex. This reflects the ideas of BNs.
>
> > Furthermore, in Eq. 2, it is not really true that "commonly, in BNs, p(xi|pa(vi))=g(dist(xi,x^i))" -- BNs aim at representing dependence distributions between variables, like "rainy weather" and "risk of aqua planning", and it's not really clear what a distance function between those variables should represent.
>
> Our scenario differs from the common applications of Bayesian networks. In our setting, $x_i$, $z_i$, and $\hat{x}_i$ are all continuous variables rather than discrete ones. For continuous random variables, when $x$ is dependent on $z$, a Gaussian distribution $\mathcal{N}(\hat{x}, \Sigma)$ is typically used to model the probability distribution of $x$. Here, $\hat{x}$ refers to the estimate of $x$ computed using $z$. For instance, as mentioned in \[1]\[2]\[3], $\hat{x}_i$ can be estimated using a linear function,while [4] suggests employing neural networks. This process is equivalent to computing the squared Mahalanobis distance between $x$ and $\hat{x}_i$ and applying it to the function $\exp(-u)$, corresponding to the functions $\text{dist}(\cdot)$ and $g(\cdot)$  in Equation 2. In this paper, we extend this approach by replacing the conventional squared Mahalanobis distance with a custom distance function defined in Equation 6, better aligned with the characteristics of our scenario.
>
> [1] Li, Chenzhao, and Sankaran Mahadevan. "Efficient approximate inference in Bayesian networks with continuous variables." Reliability Engineering & System Safety 169 (2018): 269-280.
>
> [2] Page Jr, Thomas J. "Multivariate statistics: A vector space approach." Journal of Marketing Research 21.2 (1984): 236-236.
>
> [3] Murphy, Kevin P. "A variational approximation for Bayesian networks with discrete and continuous latent variables." Proceedings of the Fifteenth conference on Uncertainty in artificial intelligence. 1999.
>
> [4] Hofmann, Reimar, and Volker Tresp. "Discovering structure in continuous variables using Bayesian networks." Advances in neural information processing systems 8 (1995).

---

> ### Author Response · Authors · 2024-11-20
> **Official Comment to Reviewer 1SZm (2/2)**
>
> > Furthermore, the paper is based on the construction of "thought graphs", but it's not really clear what these are -- are they task dependent? How do they look like? Given a taks, how does someone create a "thought graph"? The paper also says that "to facilitate computation, we further represent the vertex attribures as the BERT embedding of corresponding text" -- what is going on exactly?
>
> While the definition of thought graphs may vary across different tasks — for instance, in math reasoning tasks, the vertices of a thought graph represent the operations performed at each intermediate step, whereas in logical reasoning, they represent the intermediate conclusions — thought graphs are not task-dependent. The same code can be used to generate thought graphs for all datasets, which can then be used to retrieve in-context examples.
>
> We constructed a thought graph for each of the four datasets (GSM8K, AQUA, MBPP, and ProofWriter) to represent the reasoning process, as shown in Figure~3.
>
> As described on page 5, line 230, we first prompt the LLM to generate formalized reasoning representations (FRRs), which are then parsed using a rule-based method to obtain the thought graph. The prompt used to generate the FRRs can be found in Appendix D.1, and the pseudocode for parsing the FRRs into a thought graph is provided in Algorithm 1. Additionally, you can refer to the illustration in the lower part of Figure 2.
>
> As shown in the thought graph for the GSM8K dataset in Figure 3, when constructing the thought graph for GSM8K, each vertex corresponds to an string (e.g., "add", "multiply"),  which represents the operations taken in the reasoning process. As previously mentioned, we aim to compute the joint probability of these operations, which requires convert these strings into computable vectors. Therefore, we use BERT to convert the string corresponding to each vertex into an embedding representation.
>
> > Then, my understanding is that the BN defines a distribution over "thoguht graphs", and that the distribution over \hat x^i; what does \hat x^i represent? (E.g. in Eq. 8)
>
> As previously mentioned, in the scenario of continuous random variables, when $x$ depends on $z$, a Gaussian distribution $\mathcal{N}(\hat{x}, \Sigma)$ is typically used to model the probability distribution of $x$. In this case, $\hat{x}_i$ represents an estimate of the mean of $x$. In our scenario, $\hat{x}_i$ plays a similar role, serving as an estimate for $x_i$. Here, $x_i$ refers to the embedding of a particular string, which is used to represent the operation corresponding to each vertex in the thought graph.
>
> **Q1: Can you please explicitly state, e.g., what joint the BN is representing? What exactly is a "graph of thought"? Can you please provide a brief, clear, and self-contained description of the proposed method?**
>
> Taking the mathematical reasoning task as an example, as depicted in the GSM8K subfigure of Figure 3, a thought graph is a directed graph where each vertex $v_i$ corresponds to a vector $x_i \in \mathbb{R}^{n_f}$. Each vertex's attribute $x_i$ is the embedding of a string (e.g., "add" or "multiply") and represents the corresponding operation. Edges in the graph represent the order of computations; for instance, an edge from the vertex labeled Emb("add") to the vertex labeled Emb("multiply") signifies performing addition first and then using the result in a multiplication. A similar conceptualization is also seen in works such as GraphMR [1].
>
> Our approach is aimed at selecting in-context examples for a given query in multi-step reasoning tasks. During the preparation phase, we generate a thought graph for each candidate example to represent its reasoning process. For a given query, we first generate its thought graph and then select the candidate examples whose thought graphs are most similar to that of the query. The similarity between thought graphs is computed using a probability model inspired by BN. The method begins by estimating the parameters of the probability model from one thought graph, and then uses these parameters to compute the likelihood of generating another thought graph, reflecting the similarity between the two graphs.
>
> [1] Feng, Weijie, et al. "GraphMR: Graph neural network for mathematical reasoning." Proceedings of the 2021 Conference on Empirical Methods in Natural Language Processing. 2021.

---

> ### Author Response · Authors · 2024-11-29
> **Official Comment to Reviewer 1SZm**
>
> We would like to sincerely thank you for your time and efforts. If there are any concerns that have not been fully addressed, please let us know. We will make every effort to resolve them.

---

> > ### Comment · Reviewer_1SZm · 2024-11-29
> > **Discussion**
> >
> > > Furthermore, we sincerely apologize for the confusion caused by referring to our probabilistic model as a "Bayesian Network" in the paper. It should be noted that while our approach is inspired by BNs, it has some differences.
> >
> > But you still refer to it as a BN in the updated paper -- you just added some BN-related citations to the PDF, but all the clarity and understandability issues I mentioned are still there (e.g., that the parent relationships in a BN are "typically modeled" via a distance function)

---

> > > ### Author Response · Authors · 2024-11-30
> > > **Official Comment to Reviewer 1SZm**
> > >
> > > Thank you for your response. We will try our best to resolve the comments.
> > >
> > > First, we sincerely apologize for not directly modifying the issues you raised in the PDF due to the limited time available during the rebuttal period. However, we have provided detailed responses to each of your concerns in the comments. For example, our explanation addressing your concerns about the distance function can be found in the second part of "Official Comment to Reviewer 1SZm (1/2)". If these explanations do not fully address your concerns, we are ready to resolve any further questions you may have.
> > >
> > > Finally, as the current PDF cannot be revised, we assure that the necessary revisions and clarifications will be included in future updates to enhance the clarity and comprehensibility of the work.

---

> > > ### Author Response · Authors · 2024-12-04
> > > **Official Comment to Reviewer 1SZm**
> > >
> > > Dear Reviewer 1SZm,
> > >
> > > We would like to sincerely thank you for your time and valuable feedback. Could you kindly confirm if our responses have adequately addressed your concerns? If the concerns have been addressed, we would be grateful if you could consider updating your score accordingly. Thank you again for your thoughtful consideration.
> > >
> > > Best regards,
> > >
> > > Authors

---

### Official Review · Reviewer_GZw4 · 2024-11-03

**Soundness:** 3
**Presentation:** 3
**Contribution:** 3
**Rating:** 6
**Confidence:** 3

**Summary:**

This paper focuses on improving the selection of the in-context examples. It proposes GraphIC, which leverages the graph-structure and Bayesian Network to select in-context examples for complex reasoning tasks. Experiments on three types of reasoning tasks (math, code, and logical reasoning) demonstrate GraphIC outperforms the other ICE selection methods.

**Strengths:**

1. This paper uses a formalized reasoning representation to construct a thought graph for complex reasoning problems. Based on that, it can better model the underlying reasoning process than the semantic representation of natural language.
2. It enhances the graph embedding by the personalized PageRank and establishes a probabilistic model for the thought graph. GraphIC retrieves in-context examples by selecting top-k candidate examples that can maximize the probability of generating the correct thought graph. The probabilistic method can better capture the examples that can enhance the reasoning of the new query.
3. The paper verifies the proposed method on diverse reasoning benchmarks with multiple training-free and training-based retrieval baselines, and the results demonstrate its effectiveness.
4. It further conducts a comprehensive ablation study on each component and investigates the symmetry of different retrieval methods.

**Weaknesses:**

1. GraphIC relies on the thought graph, which is generated by formalized reasoning representation from LLMs. How to ensure the correctness of the thought graph of candidate examples? Will it be multiple possible thought graphs for the same query? Will these factors affect the robustness of GraphIC?
2. For a test query q, GraphIC first creates the thought graph G^q without the ground-truth answer and retrieve in-context examples to maximize the probability density p_i(X^q). This also assumes that the thought graph G^q is correct. What if the thought graph has an incorrect reasoning process? Will it mislead the retrieval and thus affect the performance?

**Questions:**

1. Can you estimate the extra inference and time cost for GraphIC? For other retrieval baselines, they do not need to create the thought graph.
2. In Figure 4, the performance keeps increasing with more in-context examples. Do you think it will benefit from more than even more examples? For example, on MBPP, can 16-shot outperform the CEIL baseline?

---

> ### Author Response · Authors · 2024-11-20
> **Official Comment to Reviewer GZw4 (1/2)**
>
> Thanks for the detailed comments and insightful suggestions! We will try our best to resolve the comments.
>
> **W1: GraphIC relies on the thought graph, which is generated by formalized reasoning representation from LLMs. How to ensure the correctness of the thought graph of candidate examples? Will it be multiple possible thought graphs for the same query? Will these factors affect the robustness of GraphIC?**
>
> We cannot guarantee that the thought graph generated by the LLM is entirely accurate. However, our experimental results show that the thought graph generated by the LLM is sufficiently accurate. This is because relying on the generated thought graph enables optimal performance across all datasets. Furthermore, when an LLM fails to correct construct a thought graph, it does not mean that the thought graph and the ground truth are entirely different. In fact, they share many similarities, with only a few steps differing. This also proves helpful in the retrieval process of in-context learning.
>
> We also considered that the LLM's multiple outputs could lead to different results. As described in the implementation details, the temperature of the LLM was set to 1e-5 for the generation of the thought graph. This setting ensures method stability, enabling identical results across multiple executions.
>
> **W2: For a test query q, GraphIC first creates the thought graph G^q without the ground-truth answer and retrieve in-context examples to maximize the probability density p_i(X^q). This also assumes that the thought graph G^q is correct. What if the thought graph has an incorrect reasoning process? Will it mislead the retrieval and thus affect the performance?**
>
> As you mentioned, LLMs may generate incorrect thought graphs, but this doesn’t imply that the reasoning process they represent is entirely different from the ground truth. There could be partially the same, also helps in selecting good in-context examples. In fact, many incorrect thought graphs are "almost correct", involving only minor errors [1]. This is particularly useful when selecting in-context examples.
>
> Additionally, to further investigate whether incorrect thought graphs could mislead the retrieval process, we selected a subset from each dataset, containing all queries associated with incorrect thought graphs. We evaluated GraphIC on these four subsets and compared its performance with that of top training-based (DQ-LoRe) and training-free (Complex-CoT) baselines. The results, shown in the table below, reveal that even when GraphIC uses incorrect thought graphs to retrieve in-context examples, it still achieves a significant performance advantage. Note that the performance in the table below are substantially lower than those in Table 1. This is because the queries that lead to incorrect thought graphs are typically the most difficult ones.
>
> | Model       | GSM8K     | AQUA      | MBPP      | ProofWriter |
> | ----------- | --------- | --------- | --------- | ----------- |
> | Complex-CoT | 38.58     | 28.68     | 4.06      | 54.02       |
> | DQ-LoRe     | 40.83     | **33.33** | **16.75** | 65.51       |
> | GraphIC     | **43.08** | **33.33** | 15.23     | **70.11**   |
>
> [1] Wei, Jason, et al. "Chain-of-thought prompting elicits reasoning in large language models." Advances in neural information processing systems 35 (2022): 24824-24837.

---

> ### Author Response · Authors · 2024-11-20
> **Official Comment to Reviewer GZw4 (2/2)**
>
> **Q1: Can you estimate the extra inference and time cost for GraphIC? For other retrieval baselines, they do not need to create the thought graph.**
>
> Our method belongs to the category of methods that use the generated output of LLMs to select in-context examples, which also includes methods such as Skill-kNN and DQ-LoRe. These approaches involve using the LLM during the retrieval phase, resulting in longer retrieval times compared to other baselines. However, by leveraging the power of LLMs, they are suitable for complex tasks. While Skill-kNN and DQ-LoRe do not require creating the thought graph, both models still leverage the output from LLMs to enhance retrieval, introducing an additional inference step. The time costs for the three models are presented in the table below. Specifically, the retrieval time for the GraphIC model is similar to that of DQ-LoRe, and slightly higher than Skill-kNN. Despite this, GraphIC significantly outperforms Skill-kNN in terms of performance. Moreover, compared to DQ-LoRe, which has the same retrieval time, GraphIC not only delivers superior performance but also greatly reduces both the prepare time and training time required by DQ-LoRe.
>
> |               | Skill-kNN | DQ-LoRe | GraphIC |
> | ------------- | --------- | ------- | ------- |
> | prepare time  | 37min     | 20h     | 1.5h    |
> | traning time  | -         | 16h     | -       |
> | retrieve time | 0.30s     | 0.4s    | 0.4s    |
>
> Here, "prepare time" refers to the time spent generating the necessary outputs for retrieval, such as generating the required skills for all candidate examples in Skill-kNN. For our evaluation, we used the GSM8K dataset with the LLM configured as Llama-3.1-8B-Instruct.
>
> **Q2: In Figure 4, the performance keeps increasing with more in-context examples. Do you think it will benefit from more than even more examples? For example, on MBPP, can 16-shot outperform the CEIL baseline?**
>
> In order to analyze whether GraphIC can benefit from more than even more examples, we tested the performance of GraphIC and other top-3 baselines in the 16-shot setting. The experimental results are shown in the table below. The results indicate that while the performance of other baselines declines at 16 shots, GraphIC maintains an improvement. This makes GraphIC superior to the other baselines in the 16-shot scenario.
>
> | Model | BM25 | EPR  | CEIL | GraphIC  |
> | ----- | ---- | ---- | ---- | -------- |
> | MBPP  | 60   | 60.6 | 60.6 | **62.6** |

---

> > ### Comment · Reviewer_GZw4 · 2024-11-25
> >
> > Thanks for the authors' effort to address my concern. After carefully reading the response and the other reviews, I would like to keep my score as it is.
> >
> > The main concern still lies in the correctness of the thought graph. There is no guarantee (or verification) of correctness, and the incorrect thought graph will result in poor performance (as shown in AQUA and MBPP).
> > I also have some doubts about the claim that *incorrect thought graphs are "almost correct"*, which may not hold for complex and difficult problems.

---

> > > ### Author Response · Authors · 2024-11-25
> > > **Official Comment to Reviewer GZw4**
> > >
> > > We sincerely appreciate your feedback and will make every effort to address any concerns you might still have.
> > >
> > > > There is no guarantee (or verification) of correctness, and the incorrect thought graph will result in poor performance (as shown in AQUA and MBPP).
> > >
> > > We argue that in our scenario, it is more appropriate not to guarantee the correctness of thought graphs for the following reasons:
> > >
> > > 1. **Verifying the correctness of thought graphs incurs a significant computational cost.** For instance, this may involve multiple queries to the LLM with majority voting or interactions with a theorem prover (e.g., Isabelle), which is unacceptable in practical scenarios. And GraphIC has demonstrated sufficiently strong performance, which does not need to compromise computation time for additional improvements.
> > > 2. **Ensuring the correctness of the thought graph is essentially equivalent to solving the query, which contradicts the purpose of our approach.** If we can correctly generate a thought graph for a question, it means we have almost solved the problem. In this scenario, using the thought graph to prompt the LLM might be more effective than employing it to retrieve in-context examples.
> > > 3. **While the thought graph may be incorrect, it remains effective in helping identify high-quality in-context examples.** As shown in the table in the response to W2, it should be noted that the table displays a subset of queries corresponding to all incorrect thought graphs. Therefore, these queries correspond to the most challenging cases, where **all methods demonstrated poor performance**. Therefore, relative values are more insightful than absolute ones. Notably, GraphIC continues to exhibit superior performance even on this challenging subset.
> > >
> > >    Additionally, it should be noted that this subset contains queries that cause GraphIC to make mistakes. As a result, **the comparison may not be entirely fair and tends to disadvantage GraphIC**. We present this experimental result to show that even when GraphIC makes mistakes in generating thought graphs, it still performs comparably to the second-best baseline on AQUA and MBPP, and outperforms on GSM8K and ProofWriter. Therefore, slightly lagging behind other baselines on this subset does not indicate that GraphIC demonstrates poor performance.
> > > 4. **Similar methods, such as Skill-kNN and DQ-LoRe, also do not guarantee the correctness of LLM outputs, which have shown good results.**
> > >
> > > > I also have some doubts about the claim that *incorrect thought graphs are "almost correct"*, which may not hold for complex and difficult problems.
> > >
> > > As you mentioned, for more complex tasks, incorrect thought graphs are not necessarily "almost correct," but they are also unlikely to be "completely wrong". Partial correctness can still provide valuable support for retrieving in-context examples.
> > >
> > > To validate this, we compared GraphIC against the top-performing training-based and training-free baselines (DQ-LoRe and Complex-CoT) on the complex and difficult MATH \[1] dataset. The results, displayed in the table below, demonstrate that GraphIC achieves superior performance, even on very complex and difficult tasks.
> > >
> > > |      | Complex-CoT | DQ-LoRe | GraphIC  |
> > > | ---- | ----------- | ------- | -------- |
> > > | MATH | 18.8        | 20.4    | **21.8** |
> > >
> > > Due to the time limit during the rebuttal period, we randomly selected 500 samples from the training and testing data categorized as "level 5" difficulty in MATH, which were used as the candidate set and test set.
> > >
> > > [1] https://github.com/hendrycks/math/

---

> > > ### Author Response · Authors · 2024-12-01
> > > **Official Comment by Reviewer GZw4**
> > >
> > > We would like to sincerely thank you for your time and efforts. We kindly ask whether our responses have addressed your concerns. If there are any concerns that have not been fully addressed, please let us know. We will make every effort to resolve them.

---

> > > ### Author Response · Authors · 2024-12-02
> > > **Official Comment to Reviewer GZw4**
> > >
> > > Dear Reviewer GZw4,
> > >
> > > We would like to sincerely thank you for your time and valuable feedback. Could you kindly confirm if our responses have adequately addressed your concerns? If there are any remaining issues, we would greatly appreciate it if you could point them out so we can make further improvements. If the concerns have been fully addressed, we would be grateful if you could consider updating your score accordingly. Thank you again for your thoughtful consideration.
> > >
> > > Best regards,
> > >
> > > Authors

---

> > > ### Author Response · Authors · 2024-12-04
> > > **Official Comment by Reviewer GZw4**
> > >
> > > Dear Reviewer GZw4,
> > >
> > > We would like to sincerely thank you for your time and valuable feedback. Could you kindly confirm if our responses have adequately addressed your concerns? If the concerns have been addressed, we would be grateful if you could consider updating your score accordingly. Thank you again for your thoughtful consideration.
> > >
> > > Best regards,
> > >
> > > Authors

---

### Meta-Review · Area_Chair_sg97 · 2024-12-21

**Metareview:**

Summary: The paper presents GraphIC, an approach for selecting examples for in-context learning (ICL) that focuses on capturing reasoning structure rather than just semantic similarity. The proposed method leverages graph-based representations called "thought graphs", something inspired by Bayesian Networks (BNs) and personalized PageRank to improve relevance scoring. Empirical studies conducted show better results for GraphIC on 4 tasks (spanning mathematical reasoning, code generation, and logical reasoning) against chosen training-free and training-based baselines.

Strengths:
- Important problem of example selection for ICL
- Good direction of using graph representation (but details are confusing as mentioned in weakness)
- Comparison with both training-free and training-based baselines (but gains are somewhat marginal currently, should show on more complex tasks where thought graphs have potential to shine brighter)
- Provides ablation for different components and robustness to incorrect thought graphs

Weakness:
- The submission received a lukewarm response from the reviewers and several concerns were raised.
- Lack of clarity in explaining key concepts like thought graphs and their construction
- Ambiguity in the proposed probabilistic framework as the joint probability distribution modelled by BNs was unclear
- Applicability to complex tasks where thought graph is motivated to shine most is not clear as there is limited evaluation on more complex datasets
- Concerns about the consistency and correctness of LLM-generated thought graphs
- Presentation issues making technical details hard to follow
- Additional inference time compute needed due to thought graph construction for marginal improvements.

Decision:
While the work shows promise and addresses an important problem, unfortunately, the paper can't be accepted in its current form and addressing all the concerns would warrant another round of reviewing.

**Additional Comments On Reviewer Discussion:**

We thank the authors and reviewers for engaging during the discussion phase towards improving the paper. Below are some of the highlights:

1. Thought Graph Definition:
- Multiple reviewers questioned the definition and mechanics (e.g. what are node, vertex, distribution, etc) of thought graphs
- Authors attempted to clarify them for 3 reviewers, but with limited success. AC's suggestion: In future version, maybe authors should present it as an abstract concept mechanically first and then show its possible interpretations.

2. Bayesian Network (BN) connection:
- Reviewers found explanation of BN modeling unclear
- Authors provided their explanations of probabilistic modeling approach
- Reviewers remain confused

3. Questions about computation time compared to baselines
- Authors provided detailed timing analysis
- Clarified different processing stages

4. Applicability to more complex tasks:
- Concerns about complex problems
- Authors added experiments on MATH dataset
- However, the use of non-standard subset makes it hard to understand the significance of the result

In summary, though the authors provided some clarifications in rebuttal, many suggested improvements (e.g. handling BN connection) would require substantial revision beyond what's possible in the rebuttal period.

---

### Decision · Program_Chairs · 2025-01-22

Reject